# Glucose and trehalose metabolism through the cyclic pentose phosphate pathway shapes pathogen resistance and host protection in *Drosophila*

**Michalina Kazek[1‡], Lenka Chodáková[1‡], Katharina Lehr[1], Lukáš Strych[1], Pavla Nedbalová[1], Ellen McMullen[1], Adam Bajgar[1], Stanislav Opekar[2], Petr Šimek[2], Martin Moos[2], Tomáš Doležal[1]***

1 Department of molecular biology and genetics, Faculty of Science, University of South Bohemia, České Budějovice, Czech Republic, 2 Laboratory of Analytical Biochemistry and Metabolomics, Institute of Entomology, Biology Centre, Czech Academy of Sciences, České Budějovice, Czech Republic

‡ These authors share first authorship on this work.
* tomas.dolezal@prf.jcu.cz

**Data Availability Statement:** All relevant data are within the paper and its Supporting Information files. Raw data for S1 Table were deposited to The

## Abstract

Activation of immune cells requires the remodeling of cell metabolism in order to support immune function. We study these metabolic changes through the infection of *Drosophila* larvae by parasitoid wasp. The parasitoid egg is neutralized by differentiating lamellocytes, which encapsulate the egg. A melanization cascade is initiated, producing toxic molecules to destroy the egg while the capsule also protects the host from the toxic reaction. We combined transcriptomics and metabolomics, including $^{13}$C-labeled glucose and trehalose tracing, as well as genetic manipulation of sugar metabolism to study changes in metabolism, specifically in *Drosophila* hemocytes. We found that hemocytes increase the expression of several carbohydrate transporters and accordingly uptake more sugar during infection. These carbohydrates are metabolized by increased glycolysis, associated with lactate production, and cyclic pentose phosphate pathway (PPP), in which glucose-6-phosphate is re-oxidized to maximize NADPH yield. Oxidative PPP is required for lamellocyte differentiation and resistance, as is systemic trehalose metabolism. In addition, fully differentiated lamellocytes use a cytoplasmic form of trehalase to cleave trehalose to glucose and fuel cyclic PPP. Intracellular trehalose metabolism is not required for lamellocyte differentiation, but its down-regulation elevates levels of reactive oxygen species, associated with increased resistance and reduced fitness. Our results suggest that sugar metabolism, and specifically cyclic PPP, within immune cells is important not only to fight infection but also to protect the host from its own immune response and for ensuring fitness of the survivor.

## Introduction

When activated by a challenge, immune cells must rapidly undertake new functions, facilitated by the rapid generation of energy and biosynthetic intermediates [1]. The majority of

European Nucleotide Archive under Study accession number: PRJEB74490 (secondary acc: ERP159178) and are availbale at: https://www.ebi.ac.uk/ena/browser/view/PRJEB74490 Raw data for S2 Table are available at: https://doi.org/10.6084/m9.figshare.25525657.v1.

**Funding:** TD received funding from the Grant Agency of the Czech Republic - Project 20-09103S; www.gacr.cz. MK received funding from the European Union's Horizon 2020 research and innovation programme under the Marie Skłodowska-Curie grant agreement No 867430 (IMMUNETREH). Funders did not play any role in the study design, data collection and analysis, decision to publish, or preparation of the manuscript.

**Competing interests:** The authors have declared that no competing interests exist.

**Abbreviations:** AGC, automatic gain control; G6P, glucose-6-phosphate; hpi, hours post infection; IT, injection time; PPP, pentose phosphate pathway; ROS, reactive oxygen species; TCA, tricarboxylic acid; TPM, transcripts per million.

mammalian immune cells and insect immune cells become more dependent on glucose and have an increased rate of glycolysis during infection [1,2]. This allows for the rapid generation of ATP and branching into other metabolic pathways, for example, the pentose phosphate pathway (PPP) [3]. We have shown that during infection of *Drosophila melanogaster* larvae by a parasitoid wasp, immune cells (hemocytes) consume more than double the amount of total systemic glucose than hemocytes in uninfected larvae [4]. Hemocytes release adenosine to suppress carbohydrate consumption by nonimmune tissues to ensure their own supply, which is critical for an effective immune response [4].

Trehalose is the primary carbohydrate of insects; when cleaved by trehalase (Treh), it provides a rapid source of glucose [5]. Unlike glucose, trehalose is a nonreducing sugar and thus the hemolymph of *Drosophila* larvae contains a tenfold higher concentration of trehalose than glucose [4,6]. This serves as a buffer for glucose homeostasis to ensure robust and stable development [7]. We previously found that hemocytes strongly up-regulate expression of the trehalose transporter Tret1-1 and Treh during immune response [4]. This was later confirmed by single cell transcriptomics studies [8,9], which showed lamellocyte-specific expression of Tret1-1 and Treh. These results suggest an important role for trehalose during immune response, which has also been shown in house flies [10].

We use a model of infection where the parasitoid wasp *Leptopilina boulardi* injects its egg into developing *Drosophila* larvae [11]. Within a few hours, the egg is recognized by circulating hemocytes (plasmatocytes), which induce an immune response. Sessile hemocytes enter the circulation and some attach to the egg while others differentiate into the lamellocytes, large flat cells which later encapsulate the egg. A melanization cascade is initiated in the forming capsule [12] and within approximately 48 h the parasitoid egg is destroyed. If the immune response is slow or inefficient, the parasitoid larva emerges from the egg and eventually consumes the host fly at the pupal stage [4].

Melanization/encapsulation serves the dual purpose of producing and concentrating toxic substances inside the capsule to kill the parasitoid while protecting the host from toxic radicals [12]. Therefore, there are metabolic requirements associated with lamellocyte differentiation (global changes in gene expression, cytoskeletal, and membrane rearrangements, etc.), production of toxic molecules (e.g., reactive oxygen species, ROS), and host protection mechanisms (e.g., antioxidant production) [13]. Metabolic reprogramming for the individual tasks involved in killing pathogens and protecting the host is essential for the survival of the infected larva and for fitness of the surviving fly. However, the actual changes in metabolism of larval hemocytes during immune response to wasp infection have not yet been investigated.

One of the most important branches from increased glycolysis during immune response is the PPP [3], generating NADPH and pentoses as precursors for nucleotide and coenzyme synthesis [14]. NADPH is essential in activated immune cells for lipid biosynthesis, for the ROS production [15], as well as for the production of antioxidants, such as glutathione, to protect the host from excessive ROS exposure [16]. Cells can dramatically increase NADPH production through cyclic PPP, which repeatedly oxidizes glucose-6-phosphate (G6P), as recently demonstrated in neutrophils [3].

Immune cells have privileged access to nutrients during immune response [17]. We hypothesize that hemocytes secure prioritized allocation of carbohydrates using trehalose, i.e., by expressing Tret1-1 along with Treh. To test this hypothesis, we employed $^{13}$C stable isotope tracing [18] to analyze metabolic changes in hemocytes and by genetically manipulating trehalose metabolism to investigate its role during immune response to parasitoid wasps. We found that hemocytes express several carbohydrate transporters, some of which are dramatically up-regulated upon infection. $^{13}$C tracing experiments showed that activated hemocytes uptake more glucose and trehalose and increase glycolysis and cyclic PPP, all essential for lamellocyte

production and larval resistance. Systemic trehalose metabolism is important for an effective immune response, but trehalose itself is only metabolized in fully differentiated lamellocytes, which is not necessary for resistance but instead appears to be important for lamellocyte-mediated host protection.

## Results

### Activated hemocytes increase the expression of carbohydrate transporters and trehalase

Since *Drosophila* hemocytes increase their uptake of carbohydrates during parasitoid wasp infection [4], we analyzed the expression of carbohydrate transporters in the hemocytes. The SLC2 family of hexose sugar transporters in *Drosophila* comprises 31 genes (S1 File; FlyBase ID: FBgg0000691), most of which have not been functionally characterized. Our transcriptomics shows that the expression of 10 of these genes is greater than 3 transcripts per million (TPM) in hemocytes (S1 File and S1 Table) and 4 of these (*Tret1-1*, *CG4607*, *sut1*, and *CG1208*) become expressed in hemocytes during infection (Fig 1A). Tret1-1 has been characterized as a glucose and trehalose transporter [19] and CG4607 appears to be involved in lysosomal glucose metabolism [20]. Both *Tret1-1* and *CG4607* are weakly expressed in hemocytes in the uninfected state, but their expression strongly increases upon infection (Fig 1A and S1 File). Based on scRNAseq [9], both are expressed exclusively in lamellocytes (S1 File). There are 2 mRNA variants of Tret1-1, with *Tret1-1-RA* increasing 31-fold upon infection, whereas there is no increase in the *Tret1-1-RB* variant (S1 File). *sut1* is highly expressed in most hemocyte types in both uninfected and infected states (Fig 1A and S1 File). MFS3, which belongs to the SLC17 family of organic anion transporters, has also been shown to transport glucose and trehalose [21] and its high expression in uninfected hemocytes decreases during infection (Fig 1A and S1 File). The as yet functionally uncharacterized CG1208 shows the strongest increase in expression upon infection, predominantly in lamellocytes (Fig 1A and S1 File). Thus, MFS3 and sut1 may provide basal carbohydrate transport in most hemocyte types in the uninfected state, whereas Tret1-1 and presumably CG4607 and CG1208 likely enhance carbohydrate transport in lamellocytes during infection.

While *sut1* is moderately expressed in the lymph gland, we did not detect increased expression of transporters other than *CG1208* at 18 hours post infection (hpi) (Fig 1A). MFS3 and sut1 also appear to be the major carbohydrate transporters in the wing discs which, however, display no changes in carbohydrate transporter expression during infection (Fig 1A).

Notably, in addition to the transporter Tret1-1, expression of the enzyme Treh, which converts trehalose to 2 glucose molecules, is strongly up-regulated in hemocytes upon infection (Fig 1A and 1B), primarily in lamellocytes (S1 File). Since Treh occurs in 2 forms, cytoplasmic (cTreh) and secreted (sTreh; Fig 1D) [22], we used transcript-specific qPCR to distinguish between them. Multiple cTreh transcripts (*RA*, *RD*, *RG*, *RE*) increased 35-fold in hemocytes upon infection (Fig 1C), whereas those encoding sTreh (*RC*, *RF*) were induced by infection only slightly above the basal level of cTreh. Therefore, substantially more cTreh than sTreh is expressed in hemocytes during infection.

We generated a cTreh specific mutant, *Treh[RAΔG4]*, by replacing 47 bp including the first 2 start codons with the Gal4 coding sequence, which can also serve as a cTreh expression reporter (Fig 1D). The homozygous *Treh[RAΔG4]* mutant shows a phenotype similar to the cTreh-specific mutant *Treh[c1]* [22], with 20% lethality during pupal development and two-thirds of adult flies becoming lethargic and dying within 3 days of eclosion. Using a heterozygous *Treh[RAΔG4]* line crossed with flies carrying *UAS-GFP*, we found lamellocyte-specific expression of cTreh (S1 Fig) consistent with scRNAseq data. cTreh was also expressed in other

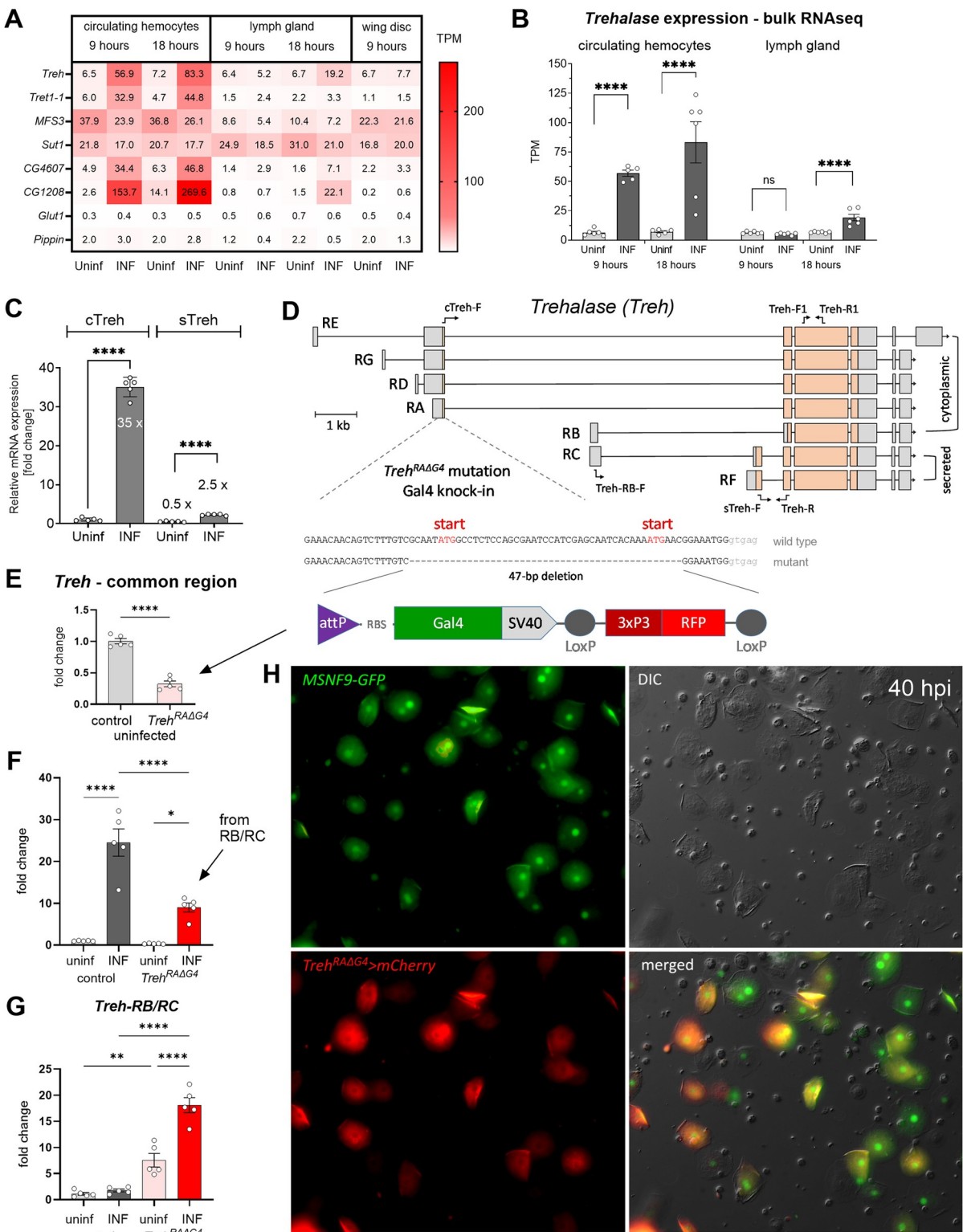

**Fig 1. Analysis of carbohydrate metabolism gene expression.** (A) Expression heat map (bulk RNAseq) of selected transporters and trehalase in circulating hemocytes, lymph gland and wing disc from uninfected (Uninf) and infected (INF) third-instar larvae collected 9 and 18 hpi (0 h = 72 h after egg laying). Mean values given in each cell are TPM—data in S1 Table. (B) Expression of the *trehalase* (*Treh*) gene (bulk RNAseq) in circulating hemocytes and lymph gland. Each dot represents a biological replicate in TPM, bars represent mean ± SEM. Samples were analyzed using DESeq2 in Geneious Prime (S1 Table), **** shows adjusted *P* < 0.0001, ns = not significant. (C) Transcript-

specific analysis of *Treh* expression by RT-qPCR 18 hpi. cTreh, represents cytoplasmic trehalase (primers cTreh-F and Treh-R shown in (D)), increases 35-fold after infection. sTreh represents secreted trehalase (primers sTreh-F and Treh-R shown in (D)). Bars show fold change compared to uninfected cTreh samples (expression levels were normalized by *RpL32* expression in each sample), each dot represents a biological replicate. An unpaired two-tailed Welch's *t* test was used to compare uninfected and infected samples; ****$P < 0.0001$ (numerical values in S1 Data). (D) Map of the *Treh* gene with individual transcripts (RA-RG). Lines indicate introns, boxes represent exons with coding sequence in orange. Labeled arrows show primers used for RT-qPCR expression analysis. Schematic representation of a Gal4 knock-in into the first exon of *Treh-RA*, creating a 47-base deletion that removes both cTreh start codons, replaced by a cassette containing the Gal4 coding sequence and an RFP marker, expressed downstream of the P3 regulatory sequence in the fly eye. The resulting fly strain is *Treh[RAΔG4]*. (E, F) General *Treh* expression analysis of hemocytes from control and homozygous *Treh[RAΔG4]* mutant by RT-qPCR 18 hpi using primers common for all transcripts (Treh-F1 and Treh-R1 shown in (D)). (G) Treh-RB and RC transcript specific expression analysis of hemocytes from control and *Treh[RAΔG4]* mutant by RT-qPCR 18 hpi using primers specific to Treh-RB/RC (Treh-RB-F and Treh-R shown in (D)). (E–G) Bars show fold change compared to uninfected control samples (expression levels were normalized by *RpL32* expression in each sample), each dot represents a biological replicate. An unpaired two-tailed Welch's *t* test was used to compare samples; *$P < 0.05$, **$P < 0.01$, ****$P < 0.0001$ (numerical values in S1 Data). (H) Analysis of *cTreh* expression in hemocytes at 40 hpi. *Treh[RAΔG4]*, expressing Gal4 in the cTreh expression pattern, drives UAS-mCherry expression (red) together with a lamellocyte-specific marker MSNF9-GFP (green) in differentiated lamellocytes but not in plasmatocytes. hpi, hours post infection; PPP, pentose phosphate pathway.

larval tissues, such as imaginal discs (S1 Fig) and brain (S2 Fig), with and without infection. To verify cTreh expression in lamellocytes, we combined *Treh[RAΔG4]* driven *UAS-mCherry* with a lamellocyte-specific marker *MSNF9-GFP* (Figs 1H and S3) and found that cTreh is indeed specifically expressed in fully differentiated lamellocytes. While the expression of MSNF9 is already visible at 18 hpi in not yet fully differentiated lamellocytes (S3 Fig), cTreh starts to appear at 24 hpi (S3 Fig) and is expressed in most lamellocytes at 40 hpi (Fig 1H), including in those encapsulating the parasitoid egg, which is already being melanized (S3 Fig).

Since the increased *Treh* expression during infection is mainly owing to the *Treh-RA* transcript, we expected that this increase would be abolished in the homozygous *Treh[RAΔG4]* mutant. The overall expression of *Treh* was reduced to one-third in uninfected *Treh[RAΔG4]* homozygotes (Fig 1E), but was still up-regulated nine-fold in the infected mutant (compared to a 25-fold increase in the wild type; Fig 1F). Using transcript-specific qPCR, we found that there was a compensatory expression from the transcription start site common to both *Treh-RB* (cTreh) and *Treh-RC* (sTreh) transcripts, which are weakly expressed in wild-type hemocytes (Fig 1G). Thus, while there is a strong increase in cTreh expression in lamellocytes during infection from the *Treh-RA* (or *RD*, *RG*, *RE*) transcripts, loss of this transcript is compensated by the expression of *Treh-RB*.

Our expression analysis showed that hemocytes express multiple carbohydrate transporters, some of which are dramatically induced during infection. This is consistent with our previous results showing enhanced sugar consumption under these conditions. The combination of highly elevated expression of trehalose transporters and of the cytoplasmic form of Treh exclusively in lamellocytes, together with the compensatory expression of cTreh, suggests the importance of trehalose metabolism in these cells.

Our bulk transcriptomic analysis showed that all enzymes associated with glycolysis and PPP are strongly expressed in hemocytes in both the uninfected and infected states (Fig 2 and S2 File and S1 Table). The levels of specific mRNAs in each pathway correspond well with scRNAseq analyses (S2 File; [8,9]). The same glycolytic and PPP genes are similarly expressed in the lymph gland and wing disc with or without infection (S2 File and S1 Table). Combining our bulk transcriptomics and scRNAseq data revealed a tendency for a slight decrease in the expression of glycolytic genes in most prohemocytes and plasmatocyte-like cells during infection, while the expression shifted towards lamellocytes and crystal cells (S2 File). Expression of PPP genes was generally unchanged during infection. Phosphofructokinase (Pfk) showed the lowest expression of all glycolytic genes and was further reduced in all hemocyte types during infection, suggesting a shift away from glycolysis to PPP and back to glycolysis at the

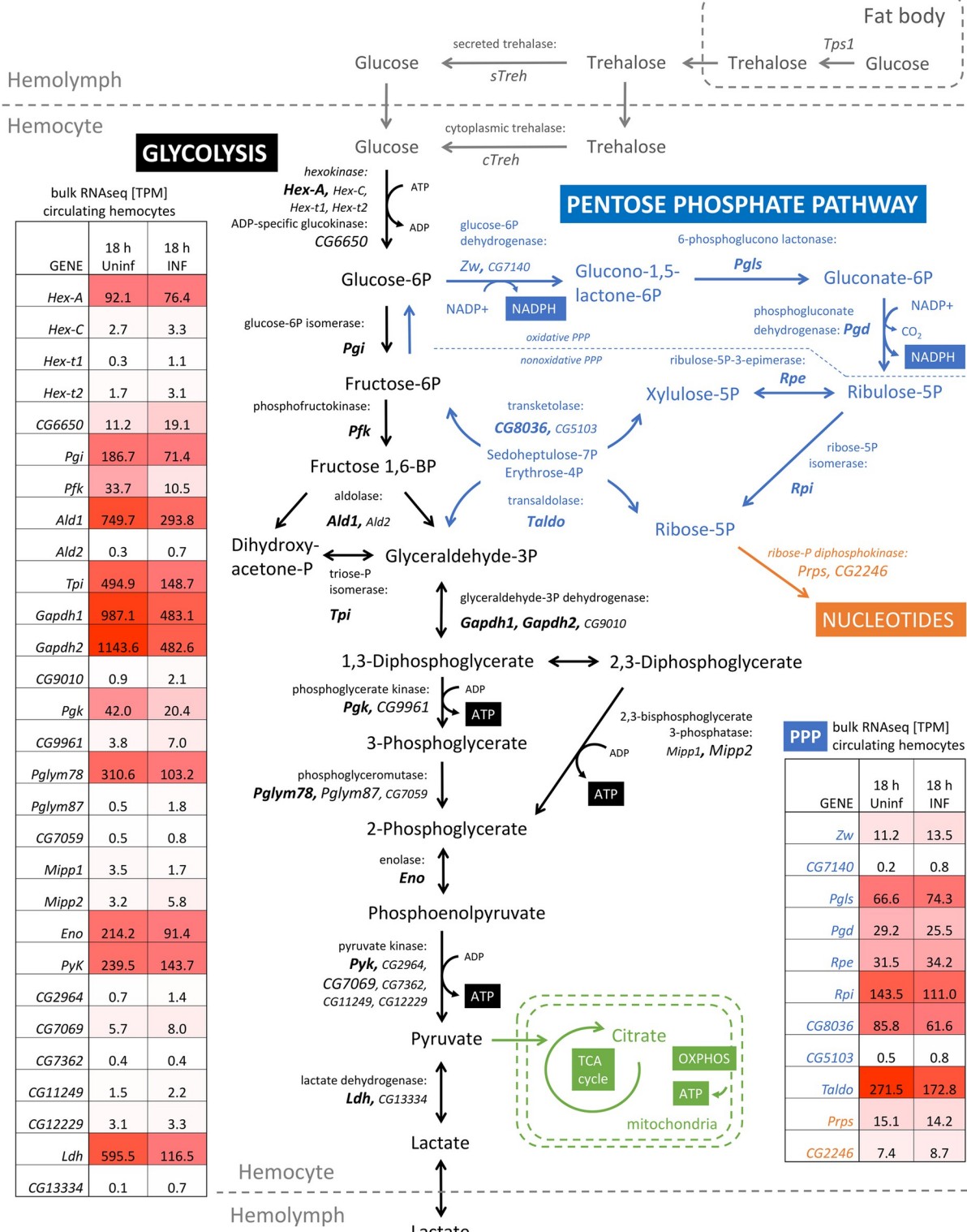

**Fig 2. Expression of glycolytic and pentose phosphate pathway enzymes in hemocytes.** Diagram showing metabolic pathways; metabolites and enzymes are abbreviated with *Drosophila* gene names; strongly expressed genes in hemocytes are in large font in bold, moderately expressed genes in large font, and insignificantly expressed genes in small font. Trehalose metabolism is shown in gray, glycolysis in black, pentose phosphate pathway in blue, purine metabolism in orange, and mitochondrial metabolism in green. Tables show glycolytic and PPP gene expression based on bulk RNAseq in TPM in circulating hemocytes at 18 hpi and from the uninfected (18 h Uninf) larvae; the shading

highlights the strength of expression—white the weakest, red the strongest—data in S1 Table. hpi, hours post infection; PPP, pentose phosphate pathway; TPM, transcripts per million.

glyceraldehyde-3P level. Overall, the expression analysis does not indicate specific changes in these metabolic pathways in either of the hemocyte types, or during infection.

Whereas in our previous study, we detected slightly increased expression (1.5- to 2-fold) of most glycolytic genes after infection using qPCR analysis [4], here we detected decreased expression of most of these genes using bulk RNAseq analysis (Fig 2). This can be explained by increased overall gene expression in activated hemocytes after infection, where normalization of RNAseq data to one million transcripts (TPM) versus normalization to a selected house-keeping gene may lead to such differences. This difference is evident, for example, when comparing the change in trehalase expression, where RNAseq shows only an 11.5-fold increase (7.2 versus 83 TPM, Fig 1B), whereas qPCR shows a 25-fold increase (Fig 1E).

## Activated hemocytes increase uptake and metabolism of glucose and trehalose via glycolysis and pentose phosphate pathway

In order to monitor changes in hemocyte metabolism during infection, we took 2 approaches based on tracing of metabolites labeled with the stable $^{13}C$ isotope: in vivo and ex vivo. In the former, we fed D-glucose-$^{13}C_6$ (fully labeled, i.e., on all 6 carbon atoms) to larvae at 16 hpi for 6 h before collecting hemocytes at 22 hpi. In addition to following metabolism in activated plasmatocytes, we assessed the last 6 h of differentiation of the first lamellocytes, which appeared complete by the end of the experiment. In the ex vivo approach, we incubated hemocytes in media containing either fully labeled D-glucose-$^{13}C_6$ or α,α–trehalose-$^{13}C_{12}$. Hemocytes collected by bleeding larvae 22 hpi were immediately incubated for 40 min with the labeled sugars. About half of the cells have already differentiated into lamellocytes. By examining only 1 time point, we are unable to determine metabolic fluxes, but we could still observe metabolic changes. It is important to note that in both approaches we worked with a heterogeneous population of hemocytes and thus observed either averaged metabolic changes that occurred across different types of hemocytes or a metabolism that occurred only in a specific hemocyte type.

After 6 h of feeding D-glucose-$^{13}C_6$, 21% of the glucose in the larval hemolymph was labeled (S4 Fig). Some of the labeled glucose was converted to trehalose in the fat body, resulting in 16% of the circulating trehalose being labeled at one of the 2 glucose monomers (S4 Fig). Both glucose and trehalose in the hemolymph increased during infection and the increase was mainly due to the unlabeled sugars (S4 Fig), suggesting a release from stores consistent with our previous results [4]. Hemocytes can directly uptake labeled dietary glucose or convert labeled trehalose to glucose. Five-fold increased content of $^{13}C$-labeled and unlabeled glucose in infection-activated relative to control hemocytes indicated that hemocytes increased sugar uptake upon infection (S4 Fig), again in agreement with our previous results [4]. A direct comparison of downstream metabolite labeling between uninfected and infected larvae is complicated by the different fraction of labeled glucose in hemocytes of uninfected and infected larvae—15% versus 7%, respectively (S4 Fig), where the increased amount of labeled glucose in infected larvae is diluted by unlabeled glucose, also taken up by the hemocytes. Nevertheless, while the m+6 fraction (with all 6 carbons labeled) of intracellular glucose after infection is half that of the uninfected fraction, the m+6 fractions of glucose-6-phosphate (G6P) are comparable and in addition there are m+2 and m+3 fractions (i.e., with 2 or 3 labeled carbons) in infected samples that are not present in the uninfected samples (S4 Fig). Thus, the overall

increase in labeled G6P indicates more intense glucose metabolism of hemocytes after infection.

Our ex vivo approach with comparable starting amount of labeled carbohydrates showed a significant decrease in the fraction of unlabeled and a significant increase in labeled glycolytic and PPP metabolites during infection, indicating increased metabolism in these pathways (Fig 3). Hemocytes of uninfected larvae metabolized glucose (Fig 3A, 3B, 3G, 3I and 3K) but little trehalose, which became metabolized substantially only during infection (e.g., Fig 3D, 3E, 3F, 3H, 3J and 3K). This is consistent with the fact that the cytoplasmic form of *Treh* is expressed only in lamellocytes. Therefore, most of the $^{13}$C incorporation from labeled trehalose can be attributed to lamellocyte metabolism. Fully labeled G6P-$^{13}C_6$ (m+6 in Fig 3D) shows increased levels upon infection, but a prominent increase was also detected in a fraction of partially labeled G6P-$^{13}C_3$ (m+3). Partially labeled G6P can arise from fully labeled glucose via cyclic PPP (Fig 3L), in which pentoses formed by oxidative PPP are converted back to G6P by the action of transketolase/transaldolase in the nonoxidative PPP and the reverse action of G6P isomerase (Figs 2, 3L, and S5; [23,24]). Since labeled pentoses represented a minor fraction (Fig 3B and 3E), they largely combined with unlabeled pentoses to form partially labeled G6P (S5 Fig). This m+3 fraction was particularly pronounced when labeled trehalose was used as the source (Fig 3D), demonstrating that cyclic PPP is active in lamellocytes. The increased m+3 fraction was also detected when labeled glucose was used, both with and without infection (Fig 3A), indicating a basal level of cyclic PPP activity in hemocytes. The m+3 fraction was also abundant in fructose-6P (Fig 3G and 3H), where it even exceeded the m+6 fraction (but note that fructose-6P could not be distinguished from glucose-1P). While m+6 fructose-6P most likely was a product of glycolysis, m+3 arose via cyclic PPP, suggesting stimulation of cyclic PPP under infection. The enhancement of cyclic PPP during infection is further supported by the in vivo experiment. Besides the expected m+6 fraction of G6P, we also detected m+2 and m+3 fractions during infection but not in uninfected larvae (S4 Fig). Before increasing amounts of labeled xylulose-5P and ribose-5P entered the transketolase/transaldolase conversion of pentoses to hexoses, G6P m+3 (pink in S5 Fig) was the main product, being formed by coupling labeled glyceraldehyde-3P with unlabeled sedoheptulose-7P. This was evident after a short ex vivo incubation (Fig 3A and 3D). Once the labeled pentoses entered the transketolase/transaldolase conversion to hexoses, G6P m+2 began to dominate (yellow in S5 Fig), as was observed after prolonged exposure to $^{13}$C in vivo (S4 Fig).

Increased labeling of ribulose-5P/ribose-5P (Fig 3B and 3E) and sedoheptulose-7P (Fig 3C and 3F), the products of PPP and transketolase/transaldolase activity, respectively, further supports enhanced PPP in hemocytes during infection; we were unable to distinguish between ribulose-5P and ribose-5P (S2 Table) and therefore present them together. Ribulose-5P and ribose-5P can arise either from oxidative or nonoxidative PPP. The opposite direction of transketolase/transaldolase activity in the latter generates pentoses without NADPH (scheme in Figs 2, 3J, and S6). While m+3 and m+5 ribulose-5P and m+5 sedoheptulose-7P can be generated by oxidative PPP, m+2 ribulose-5P, and m+4 sedoheptulose-7P are rather markers of nonoxidative PPP (Figs 3B, 3C, 3E, 3F, S4, S5 and S6). The detected mixtures of these forms indicate that hemocytes use both oxidative/cyclic PPP and nonoxidative PPP. Different hemocyte types in our heterogenous cell population likely engage these 2 branches of PPP differently.

To summarize this part, hemocytes from both uninfected and infected larvae use nonoxidative PPP to generate pentoses as well as cyclic PPP to generate NADPH. Cyclic PPP increases in hemocytes during infection and particularly in lamellocytes as evidenced by trehalose metabolism.

The m+3 fraction of glyceraldehyde-3P, which is a product of both glycolysis and PPP, increased during infection from both labeled glucose and trehalose (Fig 3G and 3H). Pyruvate

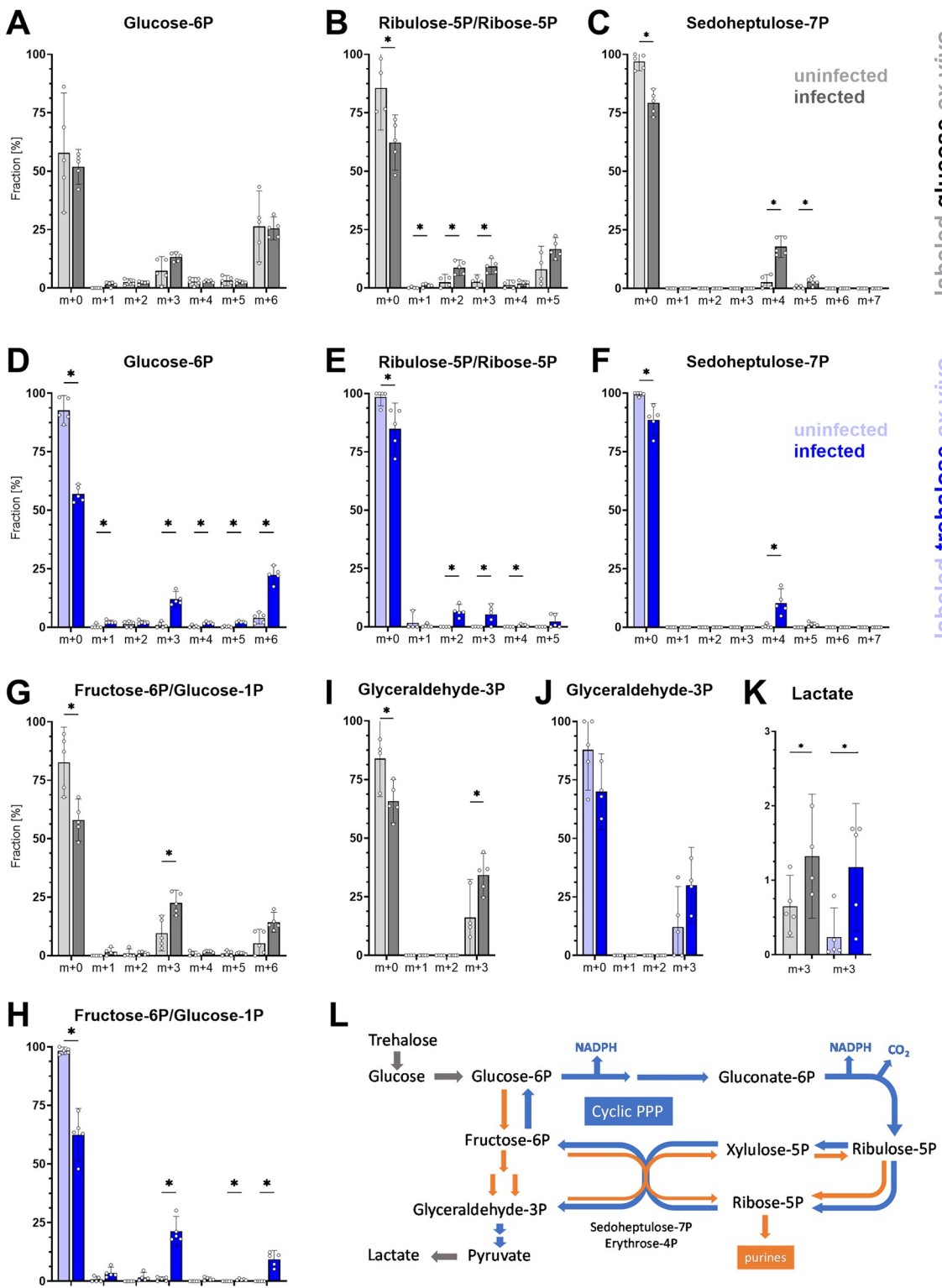

**Fig 3. Analysis of hemocyte metabolism by stable $^{13}$C isotope tracing ex vivo.** (A–K) $^{13}$C labeling of glycolytic and pentose phosphate pathway metabolites in hemocytes from uninfected (light gray/blue bars) and infected (dark gray/blue bars) larvae. Hemocytes were incubated ex vivo for 40 min in media containing 0.5 mM D-glucose-$^{13}$C$_6$ and 5 mM unlabeled trehalose (gray) or 5 mM α,α−trehalose-$^{13}$C$_{12}$ and 0.5 mM unlabeled glucose (blue). m+0 indicates the fraction of the compound with unlabeled molecular mass; m+1 indicates the fraction of the compound with one $^{13}$C-labeled carbon, and so on. Bars represent means of 5 biological

replicates with 95% CI, each dot represents 1 biological replicate; asterisks represent significant differences between uninfected and infected fractions tested by unpaired two-tailed Welch's *t* test (numerical values in S1 Data). Note that ribulose-5P and ribose-5P (B, E) and fructose-6P and glucose-1P (G, H) were indistinguishable. For lactate (K), only the m+3 fraction is shown, as m+1 and m+2 were not detected above naturally occurring levels (m+0 and m+3 make up 100%). Schematic representation (L) of the cyclic PPP in blue and nonoxidative PPP-producing pentoses for de novo purine/pyrimidine synthesis in orange. PPP, pentose phosphate pathway.

was more difficult to detect, and the labeled pyruvate reached detectable levels in the ex vivo experiment only after infection (S2 Table). We have previously shown that hemocytes release more lactate during infection [25], which is confirmed here by higher levels in hemolymph during infection (S4 Fig). The increased incorporation of $^{13}$C into lactate, from both glucose and trehalose (Fig 3I), demonstrates its increased production during infection.

Since the incorporation of $^{13}$C into tricarboxylic acid (TCA) cycle metabolites was very low (<1%) after 40 min of ex vivo incubation, we compared the incorporation into citrate from the in vivo experiment with feeding $^{13}$C-labeled glucose for 6 h. As we detected only the m+3 fraction of pyruvate (S7 Fig), we expected either m+2 (via pyruvate dehydrogenase) or m+3 (via pyruvate carboxylase) fractions of citrate. However, we detected an m+1 fraction as the highest (S7 Fig), which we also found in other TCA metabolites (S7 Fig). Therefore, we checked the $^{13}$C labeling of glutamine and glutamate as a potential source and found that it matched the fractional pattern of the TCA metabolites (S7 Fig). Since we fed larvae with labeled glucose for 6 h, some of the glucose was metabolized to amino acids; the labeled amino acids were released into the hemolymph and taken up by hemocytes as a source for mitochondrial metabolism or, via citrate, potentially for lipid synthesis. Pyruvate derived from glucose could still be used in mitochondria and contribute to the m+2/m+3 fractions, but the amino acids seem to be an important source for mitochondrial metabolism in hemocytes, both without and during infection. Nevertheless, changes in mitochondrial metabolism need to be further investigated.

Ribose-5P can be used for de novo purine/pyrimidine synthesis. This is indicated by the $^{13}$C labeling of AMP in vivo (S8 Fig). While all ribulose-5P/ribose-5P fractions in infected larvae were significantly lower (due to liberation of unlabeled endogenous glucose) than in uninfected larvae, the AMP fractions were similar, indicating increased purine production during infection. This is supported by the overall increase of AMP/ADP/ATP levels during infection (S8 Fig). Interestingly, the m+1 fraction of AMP was markedly higher compared to ribulose-5P/ribose-5P. In addition to m+1 ribose-5P as a source of m+1 AMP, labeled glycine (produced from labeled glucose similarly to glutamine; S8 Fig), added to unlabeled ribose-5P during purine synthesis, could increase the m+1 fraction of AMP. The carboxylation step utilizing labeled $CO_2$ released during oxidative PPP could be another source for the increased m+1 fraction of AMP [26].

In summary, metabolic changes occurring in hemocytes during infection include increased glucose uptake and, in lamellocytes, glucose production from trehalose. The prevailing fate of glucose during infection appears to be in cyclic PPP, where G6P is oxidized in multiple rounds, as evidenced by elevated partial G6P labeling. Increased glycolysis correlates with higher lactate release from hemocytes. A substantial portion of glucose is used for de novo purine synthesis. While the degree of glucose utilization in mitochondrial metabolism remains unclear, amino acids appear to be the source for the TCA cycle and potentially for lipid synthesis.

## Oxidative PPP is required for proper immune response

Our metabolomics analysis shows that carbohydrate uptake and oxidative PPP increase in hemocytes upon infection. To test the importance of oxidative PPP during immune response, we used a double mutant in G6P dehydrogenase (*Zwischenferment* or *Zw*) and

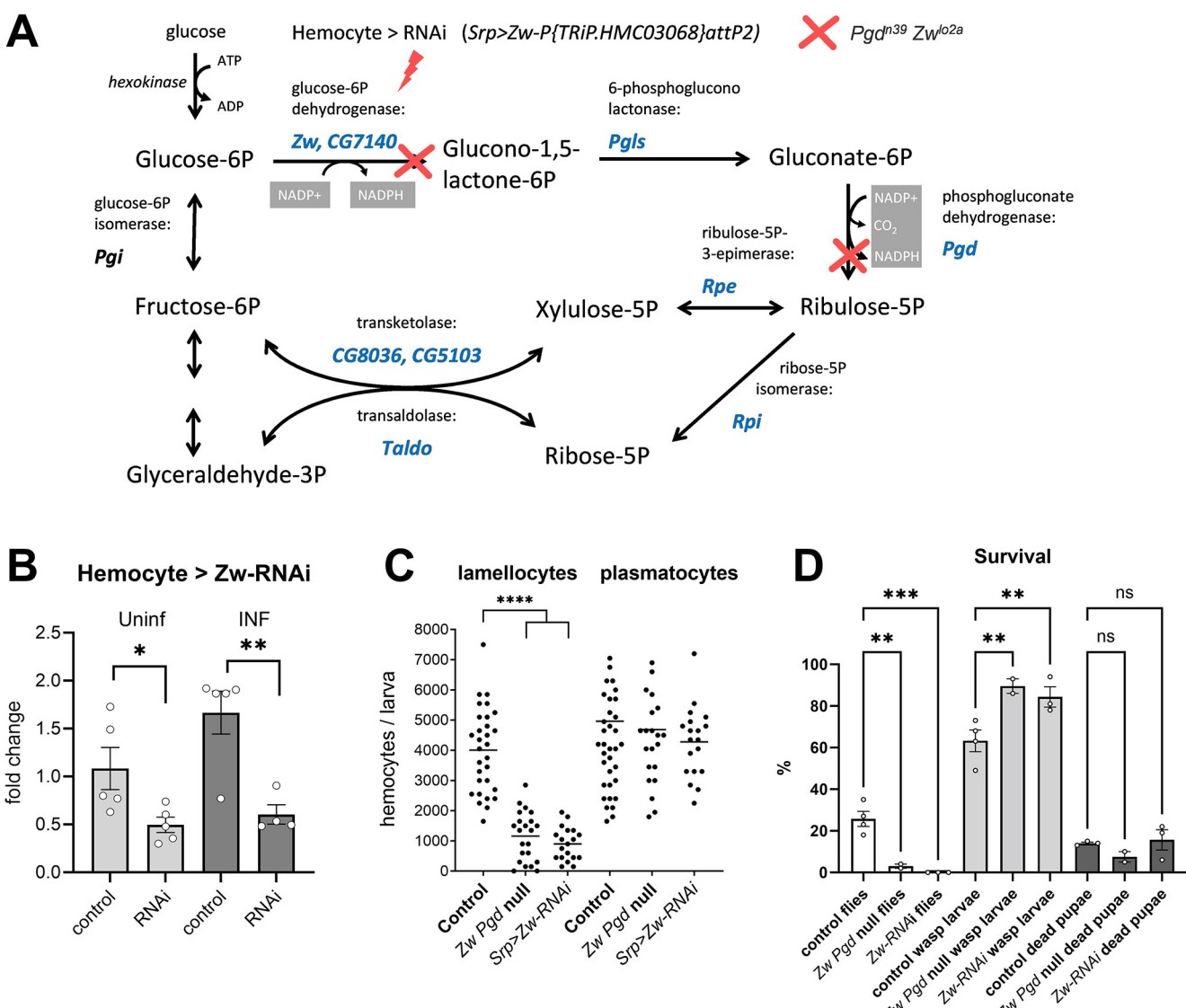

**Fig 4. Oxidative pentose phosphate pathway is important for lamellocyte differentiation and resistance.** (A) Schematic representation of pentose phosphate pathway with metabolites in black and *Drosophila* genes encoding enzymes in blue. Genetic manipulations include hemocyte-specific RNAi of *Zw* induced by Srp-Gal4 (red lightning) and double null mutant in *Zw* and *Pgd* (*Pgd[n39] pn[1] Zw[lo2a]*; red crosses). (B) Efficiency of hemocyte-specific RNAi of *Zw* tested by RT-qPCR in hemocytes from uninfected (Uninf) and infected (INF) larvae 18 hpi. Bars represent mean values of fold change compared to uninfected control samples (*Zw* expression levels normalized by *RpL32* expression in each sample); dots show individual samples, error bars represent ± SEM, the control (*Srp> P{y[+t7.7] = CaryP}attP2*) was compared to RNAi (*Srp>P{TRiP.HMC03068}attP2*) using unpaired *t* test, asterisks indicate *p* values (* $P < 0.05$, ** $P < 0.01$). (C) Number of hemocytes 22 hpi in control (*Srp> P{y[+t7.7] = CaryP}attP2*), *Zw Pgd* null mutant (*Pgd[n39] pn[1] Zw[lo2a]*), and hemocyte-specific RNAi of *Zw*. Each dot represents number of hemocytes in one larva, lines represent the mean, samples were compared by unpaired *t* test, asterisks indicate *p* value (* $P < 0.05$, **** $P < 0.0001$). (D) Survival of parasitoid wasp infection in control, null *Zw Pgd* mutant and hemocyte-specific RNAi of *Zw*. Bars show percentages of surviving flies (white), developing parasitoids (gray) and dead pupae when neither fly nor parasitoid survived (dark). Bars represent mean values; dots represent biological replicates, error bars represent ± SEM; survival rates were compared using ordinary one-way ANOVA with multiple comparisons; asterisks indicate *p* values (** $P < 0.01$, *** $P < 0.001$; numerical values in S1 Data). hpi, hours post infection.

6-phosphogluconate dehydrogenase (*Pgd*), the first and the third enzymes in oxidative PPP, respectively (Fig 4A). The *Pgd[n39] pn[1] Zw[lo2a]* mutant flies are fully viable [27]. We also used a hemocyte-specific RNAi of *Zw* (Fig 4A and 4B) using *Srp-Gal4* and the *Zw* RNAi construct under the UAS promoter (*P{TRiP.HMC03068}attP2*). The number of plasmatocytes was

altered neither in the double mutant nor by hemocyte-specific *Zw* knockdown, suggesting that basal hematopoiesis was unaffected by these manipulations (Fig 4C). However, both the double mutant and knockdown displayed decreased lamellocyte number (Fig 4C) and larval survival upon infection (Fig 4D), indicating that oxidative PPP is important for effective lamellocyte differentiation and parasitoid killing.

Oxidative PPP generates NADPH, which is important for lipid synthesis. While lipid metabolism is beyond the scope of our study, we found a rather striking difference in lipid droplets between hemocytes from control larvae and larvae with hemocyte-specific RNAi of *Zw* during infection (S9 Fig). This indicates that properly functioning oxidative PPP in hemocytes is important for their lipid metabolism during infection.

## Systemic trehalose metabolism and carbohydrates supply to hemocytes are required for efficient immune response

Metabolomics with $^{13}$C-labeled trehalose shows that trehalose is metabolized by infection-activated hemocytes. Therefore, we tested the importance of trehalose for the immune response. We first tested the hypomorphic mutation in trehalose-synthesizing gene *Tps1* (*Mi{y [+mDint2] = MIC}Tps1[MI03087] / Tps1[d2]*). *Tps1* mutant larvae develop with normal levels of glucose, glycogen, and triglycerides but their trehalose levels are only 20% those of controls [7]. Infected *Tps1* mutants differentiated significantly fewer lamellocytes than control larvae (Fig 5A). Similarly, fewer lamellocytes differentiated in null *Treh[cs1]* mutants (Fig 5B) that cannot convert accumulating trehalose to glucose [22]. We verified that hemocytes from the *Treh[cs1]* mutant were unable to metabolize $^{13}$C-labeled trehalose supplied ex vivo (Fig 5C and S2 Table). These results demonstrate that trehalose is important for efficient lamellocyte differentiation.

Next, we knocked down the *Tret1-1* transporter specifically in hemocytes (*Srp>Tret1-1-RNAi—P{TRiP.HMS02573}attP2*), which resulted in a reduction of *Tret1-1* expression upon infection to one-third compared to control (Fig 5D). Because the knockdown did not prevent the increase in *Tret1-1* expression during infection (it only partially suppressed it), ex vivo metabolism of $^{13}$C-labeled trehalose, indicated by lower incorporation into G6P (Fig 5E and S2 Table) as well as the number of lamellocytes (Fig 5F) were only slightly reduced. Since *Tret1-1* knockdown did not substantially suppress trehalose uptake, we used a null mutant of *Tret1-1* together with a null mutation of *MFS3*. The double mutant *MFS3[CRISPR] Tret1-1 [XCVI]* significantly reduced the number of lamellocytes, albeit again only slightly (Fig 5G), suggesting that other transporters ensure carbohydrate supply.

## Cell-autonomous role of trehalose in hemocytes

Null *Treh* mutation impairs systemic trehalose metabolism. Since lamellocytes specifically increase the expression of the cytoplasmic Treh, we used the specific mutations in *cTreh* to test a cell-autonomous role of trehalose metabolism in hemocytes. The *Treh[c1]* mutation (S10 Fig) should block the utilization of trehalose inside the cells but not trehalose-glucose metabolism in the circulation [22]. However, when we incubated *Treh[c1]* mutant hemocytes with $^{13}$C-labeled trehalose ex vivo, we observed no effect on trehalose metabolism in hemocytes (S10 Fig and S2 Table). The failure of the *Treh[c1]* mutation to block intracellular trehalose metabolism could be due to a second start codon in the *cTreh-RA/RD/RG/RE* transcripts (S10 Fig). However, when both start codons were removed in our *Treh[RAΔG4]* mutant (Figs 1 and S10), the homozygous mutant hemocytes still metabolized labeled trehalose almost normally (S10 Fig and S2 Table), and the mutation did not affect the number of lamellocytes (S10 Fig).

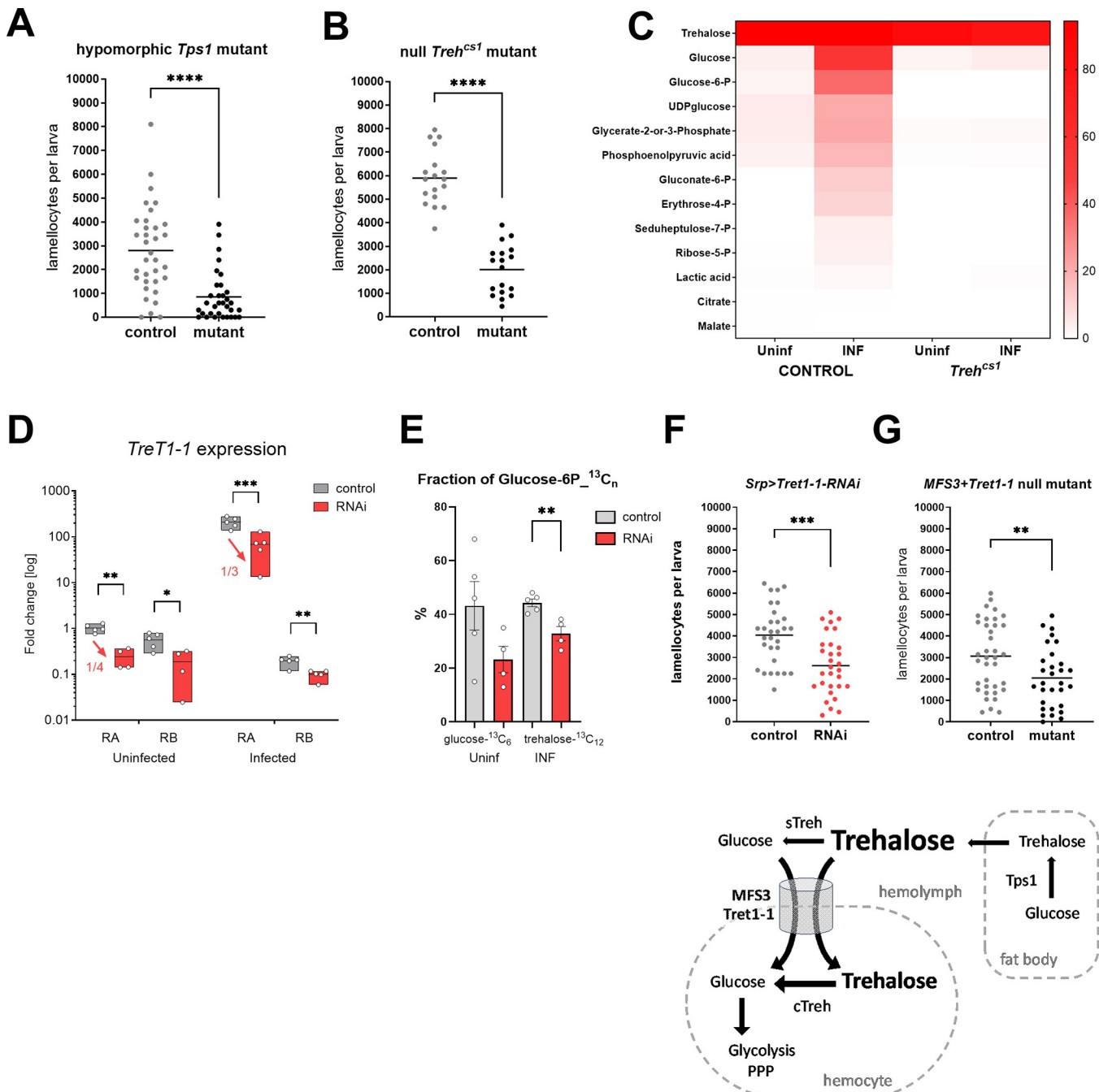

**Fig 5. Effects of systemic carbohydrate metabolism and carbohydrate supply to hemocytes on lamellocyte differentiation.** (A) Number of lamellocytes 22 hpi in control (heterozygous *Mi{y[+mDint2] = MIC}Tps1[MI03087] / +)* and hypomorphic *Tps1* mutant (*Mi{y[+mDint2] = MIC}Tps1[MI03087] / Tps1[d2]*). Each dot represents number of hemocytes per larva, lines represent mean, samples were compared by unpaired *t* test, asterisks indicate *p* value (**** $P < 0.0001$). (B) Number of lamellocytes 22 hpi in control (heterozygous *Treh[cs1] / CyO Ubi-GFP*) and null *Treh[cs1]* homozygotes. (C) Heat map of $^{13}$C-labeled fraction of metabolites from control and *Treh[cs1]* hemocytes in uninfected (Uninf) and infected (INF) conditions incubated ex vivo with α,α −trehalose-$^{13}$C$_{12}$ (data in S2 Table). (D) Efficiency of hemocyte-specific RNAi of *Tret1-1* tested by RT-qPCR in hemocytes from uninfected and infected larvae 18 hpi. Box plots (median, 75th and 25th percentiles) show fold change compared to uninfected *Tret1-1-RA* control samples (expression levels were normalized by *RpL32* expression in each sample), each dot represents a biological replicate. Gray boxes represent control (*Srp> P{y[+t7.7] = CaryP}attP2*) and red boxes *Tret1-1* RNAi (*Srp>P{TRiP.HMS02573}attP2*). Unpaired two-tailed *t* test was used to compare control with RNAi; asterisks indicate *p* values (* $P < 0.05$, ** $P < 0.01$, *** $P < 0.001$). (E) Fraction (%) of G6P-$^{13}$C$_n$ in hemocytes from uninfected and infected larvae, incubated ex vivo with labeled glucose (left bars) or trehalose (right bars), respectively. Gray bars represent mean in control (*Srp> P{y[+t7.7] = CaryP}attP2*) and red bars mean in hemocyte-specific *Tret1-1* RNAi (*Srp>P{TRiP.HMS02573}attP2*). Unpaired two-tailed *t* test was used to compare control with RNAi; asterisks indicate *p* value (** $P < 0.01$). (F) Number of lamellocytes 22 hpi in control (*Srp> P{y[+t7.7] = CaryP}attP2*) and in hemocyte-specific *Tret1-1* RNAi (*Srp>P{TRiP.HMS02573}attP2*). Each dot represents

number of lamellocytes per larva, lines represent mean, samples were compared by unpaired *t* test, asterisks indicate *p* value (*** *P* < 0.001). (G) Number of lamellocytes 22 hpi in control (*MFS3[CRISPR] Tret1-1[XCVI]* / Cyo Ubi-GFP heterozygotes) and in *MFS3 Tret1-1* double-null mutants (*MFS3[CRISPR] Tret1-1[XCVI]*). Each dot represents number of lamellocytes per larva, lines represent mean, samples were compared by unpaired *t* test, asterisks indicate *p* value (** *P* < 0.01). Numerical values for A, B, D, E, F, and G are available in S1 Data. hpi, hours post infection.

This is likely due to the compensatory expression occurring at the alternative transcription start sites common to both *Treh-RB* (cTreh) and *Treh-RC* (sTreh) transcripts (Fig 1).

Since we could not test the importance of trehalose metabolism in hemocytes with cTreh mutations, we decided to generate mitotic recombination clones [28] in the hematopoietic lineage with the *Treh[cs1]* mutation, which we verified as blocking trehalose metabolism (Fig 5C). We recombined flippase (Flp) target site *FRT42D*, *Treh[cs1]* mutation and RFP marker on the second chromosome and crossed this line to flies with *FRT42D* and the GFP marker to generate heterozygous *FRT Treh[cs1] RFP / FRT GFP* flies (Fig 6A). To induce mitotic recombination in the hematopoietic lineage of the progeny, the parental flies also carried *Srp-Gal4* and *UAS-Flp* on their third chromosomes. Mitotic recombination resulted in RFP-labeled hemocytes with a *Treh[cs1]* null mutation and their GFP-labeled wild-type siblings (Fig 6A and 6B). When mitotic clones were first induced with only RFP and GFP markers without any mutation, the expected equal ratio of RFP to GFP sibling hemocytes was detected (Fig 6C), approximately 40% each, indicating high recombination efficacy (only the remaining 20% were non-recombined GFP/RFP hemocytes). A similar result was obtained in uninfected larvae with the *Treh[cs1]* mutation, although there was a minor increase in RFP-labeled *Treh[cs1]*$^{-/-}$ hemocytes compared to the GFP-labeled wild-type hemocytes (49% versus 43%; Fig 6D). The difference is more likely to result from different genetic backgrounds of the chromosomal arms that become homozygous after recombination than from the lack of trehalase activity. A comparable result was obtained with lamellocytes from infected animals, where we found similar proportions of red and green cells—the small difference was consistent with the difference observed in uninfected larvae (Fig 6E). Generating clones with the *Treh[cs1]* null mutation did not change the total number of lamellocytes compared to larvae without induced recombination (Fig 6F). These results indicate that trehalose metabolism is dispensable for lamellocyte differentiation in a cell-autonomous manner, which is consistent with cytoplasmic trehalase not being expressed until in fully differentiated lamellocytes (Fig 1H).

To test the effect on resistance, we used the larvae bearing approximately 40% of lamellocytes mutant for *Treh*. If trehalose metabolism in the lamellocytes is necessary for parasitoid killing, we would expect lower resistance. However, induction of *Treh[cs1]* mutant clones resulted in the opposite effect, with increased resistance compared to animals with the same genetic background in which clones were not induced (S11 Fig). Although a relatively large variability was observed, the average percentage of surviving adult flies in controls was 18% and never exceeded 30%, whereas in flies with induced *Treh*$^{-/-}$ clones the average was 35% and reached up to 64% (Fig 6G). Subsequently, the percentage of surviving parasitoid wasps decreased from 65% in controls to 35% when *Treh*$^{-/-}$ clones were induced (Fig 6G).

To kill the wasp egg, the larval immune cells generate ROS [29]. However ROS are potentially harmful also to the host, which prevents damage to its own tissues by producing antioxidants [13]. Flies use the thioredoxin system instead of glutathione reductase to reduce GSSG [30], and hemocytes highly express thioredoxin reductase Trxr1 and thioredoxin Trx2 with greater expression in lamellocytes (S12 Fig). Trx2 is also a substrate for peroxiredoxins [31], which are highly expressed in hemocytes as well (S12 Fig). The thioredoxin system requires NADPH (S12 Fig), which is produced by cyclic PPP. We found increased cyclic PPP activity and NADPH levels in hemocytes, including lamellocytes, during infection (Fig 6J). We also

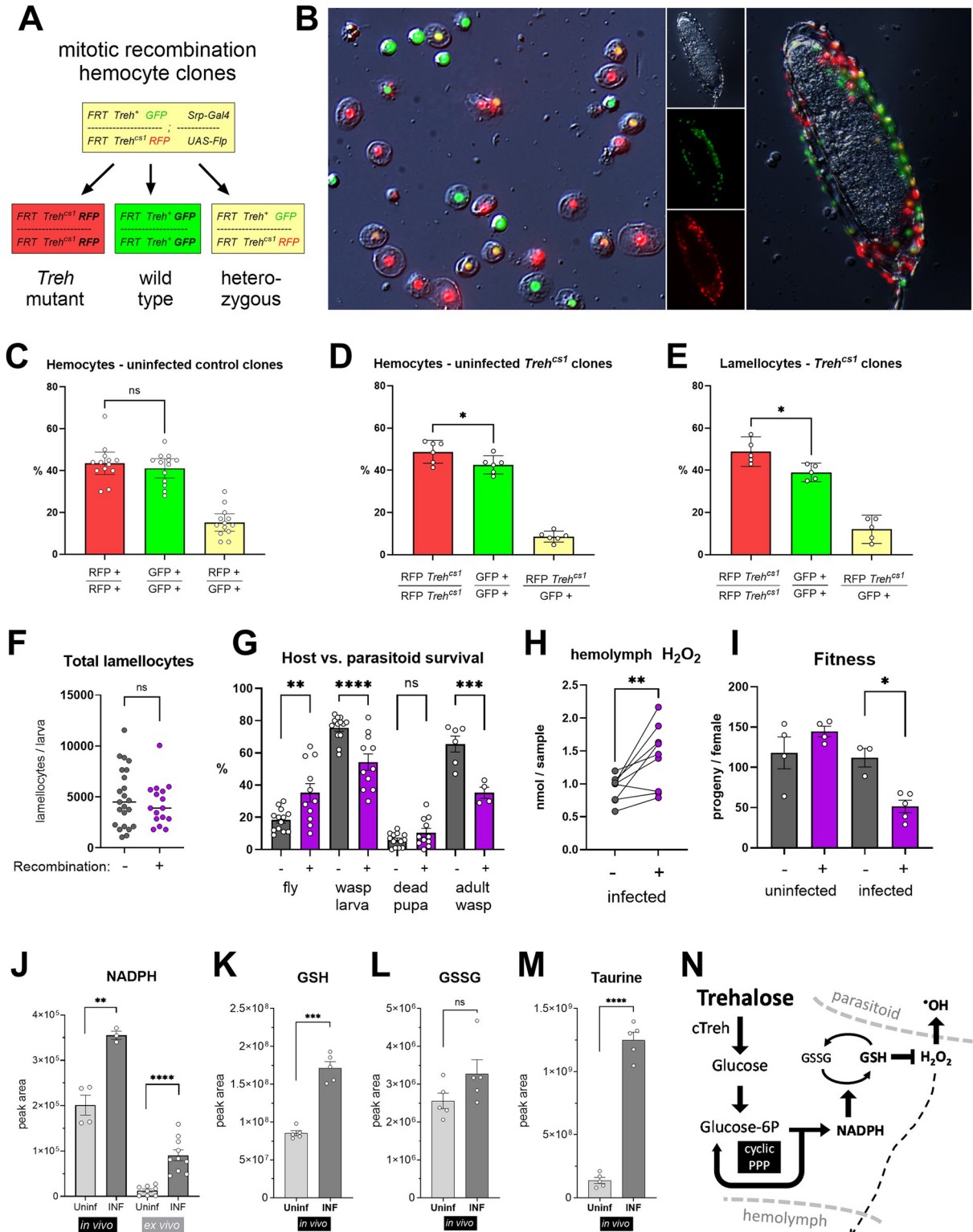

**Fig 6. Cell-autonomous role of trehalose metabolism in hemocytes.** (A) Generation of hemocyte clones with null *Treh[cs1]* mutation by *Srp-Gal4 UAS-Flp* induced mitotic recombination. Genotypes of parental and daughter cells after mitotic recombination are color marked red: RFP-marked *Treh[cs1]* mutant cells; green: GFP-marked wild-type sister clone; yellow: RFP/GFP nonrecombinant heterozygous cells. (B) Red-, green-, and yellow-marked hemocytes (both plasmatocytes and lamellocytes) upon mitotic recombination from infected larvae in circulation (left) and attached to a wasp egg (right). (C) Percentage of wild-type recombinant and nonrecombinant hemocytes in uninfected larvae

without any mutation. (D) Percentage of *Treh[cs1]* homozygous, sister wild-type and nonrecombinant heterozygous hemocytes in uninfected larvae. (E) Percentage of *Treh[cs1]* homozygous, sister wild-type and nonrecombinant heterozygous lamellocytes in infected larvae. (C–E) Bars represent mean percentage, dots represent counts from single larvae, error bars represent ± SEM; samples were compared using unpaired two-tailed *t* test; asterisks indicate *p* value (* $P < 0.05$; ns = not significant). (F–I) Controls without recombination (gray) compared to larvae with *Treh[cs1]* mutant hemocyte clones (*FRT42D GFP / FRT42D Treh[cs1] RFP; Srp-Gal4 / UAS-Flp;* purple). (F) Number of lamellocytes 22 hpi. Each dot represents number of lamellocytes per larva, lines represent mean, samples were compared by unpaired *t* test, no significant difference. (G) Percentages of surviving adult flies, wasp larvae, dead pupae (neither fly nor wasp survived), and adult parasitoid wasps. Bars represent mean percentage, dots represent biological replicates (individual infection experiments), error bars represent ± SEM, samples were compared by ordinary one-way ANOVA with Šídák's test for multiple comparisons, asterisks indicate *p* values (* $P < 0.05$; ** $P < 0.01$; ***$P < 0.001$; ns = not significant). (H) $H_2O_2$ levels in the hemolymph of infected control larvae (gray) and larvae with *Treh[cs1]* mutant clones (purple) at 30 hpi. Dots represent paired biological replicates (infection performed and $H_2O_2$ measured in parallel for 1 control and 1 mutant sample); data were compared using two-tailed paired *t* test, asterisks indicate $P < 0.01$. (I) The average number of progeny produced by an uninfected female and a female surviving the larval infection. Control (gray) and females bearing *Treh[cs1]* mutant clones (purple) were compared by two-tailed Welch's *t* test; asterisk indicates $P < 0.05$. Bars represent the average numbers of progeny produced per female, each dot represents 1 infection experiment with at least 10 surviving females, error bars represent ± SEM. (J–M) Levels of unlabeled NADPH (J), GSH (K), GSSG (L), and taurine (M) in hemocytes from experiments ex vivo or in vivo. The bars show the mean metabolite amounts expressed by the normalized peak area, samples were obtained from hemocytes from uninfected (Uninf) or infected (INF) larvae, and were compared by two-tailed Welch's *t* test; asterisks indicate *p* values (** $P < 0.01$, ***$P < 0.001$, ****$P < 0.001$; ns = not significant). (N) Model for the role of trehalose in producing antioxidants during response to parasitoid infection. Numerical values for C–M are available in S1 Data. hpi, hours post infection.

observed increased levels of the reduced glutathione form (GSH, Fig 6K) in hemocytes from infected larvae, whereas the oxidized GSSG form remained at comparable levels (Fig 6L). Besides GSH, infection elevated another antioxidant, taurine, in hemocytes (Fig 6M). Therefore, besides eliminating the pathogen, hemocytes seem to protect the host by mounting an antioxidant response.

Since trehalose metabolism in hemocytes did not appear to be important for resistance, we speculated that it may rather mediate the antioxidant host protection against own immune response. This would be consistent with trehalose being metabolized by cyclic PPP to produce NADPH, which is required for antioxidant production. Removing this activity from larval lamellocytes could decrease their ROS scavenging capacity and thus increase toxicity and resistance towards the pathogen. Therefore, we analyzed the effect of trehalase mutant hemocytes on the level of hydrogen peroxide in the hemolymph. The melanization response is generally associated with strong catalase activity in insect hemolymph, making it difficult to detect increased ROS during infection without catalase inhibitors [32]. A small increase in $H_2O_2$ was detected only at 30 hpi, but not at 20 or 40 hpi during the *Drosophila* response to parasitoid [33]. We did not use antioxidant inhibitors because we were interested in differences in antioxidant activities, and this is likely why we failed to detect a difference in hemolymph $H_2O_2$ between infected and uninfected larvae (S13 Fig). Induction of mitotic *Treh[cs1]* mutant clones did not alter $H_2O_2$ levels in uninfected larvae (S13 Fig). However, we detected a consistent, significant increase in $H_2O_2$ levels in the hemolymph of larvae with induced clones at 30 hpi (Fig 6H). This result supports the notion that trehalose may play a role in mechanisms that protect the host from oxidative stress. Down-regulation of trehalose metabolism in hemocytes leading to increased ROS may explain the observed increased resistance (Fig 6N).

There is a small, statistically insignificant increase in pupal lethality of infected larvae with induced *Treh$^{-/-}$* clones (Fig 6G), suggesting that the increased ROS levels were not lethal to the host. To explore further, we looked at the lifespan of adult flies that survived the infection. Male lifespan was comparable (S13 Fig); however, a greater number of females with induced clones died earlier than controls, with the median survival significantly reduced from 48 to 34 days, although the maximum lifespan was comparable (S13 Fig). A more pronounced effect was observed in the production of viable offspring: females with induced clones that survived infection produced less than half of progeny of controls, whereas no such difference was observed in the absence of infection (Fig 6I). Thus, induction of *Treh[cs1]* mutant clones in

hemocytes during parasitoid infection of larvae results in reduced fitness of survivors. Whether there is a relationship between increased hemolymph $H_2O_2$ and the viability of future offspring of the infected larvae remains to be tested.

We attempted to verify the effect of *Treh[cs1]* hemocytes mutant clones via another trehalase manipulation. The *Treh[cs1]* null mutation affects systemic trehalose/glucose metabolism and homozygous cTreh-specific *Treh[RAΔG4]* loss is compensated by expression of alternative transcripts. Therefore, we used the heterozygous *Treh[RAΔG4]* allele that (1) partially reduces cTreh expression (S14 Fig); and (2) expresses Gal4 in the cTreh pattern, including a strong up-regulation in lamellocytes. To suppress *Treh* in the cTreh expression pattern, we combined the Gal4 driver within *Treh[RAΔG4]* with either of 2 UAS-Treh-RNAi constructs, *Treh[HMC03381]* or *P{GD5118}v30731* (S14 Fig). Both RNAi lines recapitulated the cTreh null mutant phenotype (10% lethality in pupae and lethargic adults compared to 20% lethality in pupae and dying young adults in the null mutant). In both cases, larvae with Treh RNAi showed significantly increased resistance by killing parasitoids more often than control larvae (S14 Fig), reproducing the results obtained with *Treh[-/-]* mutant clones. However, in the case of RNAi, the increased resistance did not result in improved survival, but was associated with increased host lethality (S14 Fig). The effect of RNAi might be more severe as, unlike in *Treh[-/-]* hemocyte clones, *Treh* depletion was not limited to hemocytes but affected all tissues where cTreh is expressed and the *Treh[RAΔG4]* driver is active. Nevertheless, the increased pathogen killing at the expense of own survival suggests that trehalose may indeed play a role in host protection.

In summary, we used *Treh[-/-]* mitotic clones to test for a cell-autonomous role of trehalose metabolism in hemocytes and found that it is not required for lamellocyte differentiation or host resistance. Elimination of the ability to metabolize trehalose in 40% of lamellocytes enhanced resistance, which was associated with increased hemolymph ROS levels and reduced fitness of survivors. A similar increased in resistance was achieved with 2 independent *Treh* RNAi lines, but this was associated with higher host lethality. Since hemocytes metabolize trehalose via cyclic PPP, generating NADPH and antioxidants, these results suggest that trehalose may play a role in protecting the host from ROS that serve to kill the pathogen.

## Discussion

We previously demonstrated a systemic metabolic switch during infection of *Drosophila* larvae by parasitoid wasps, when sugar consumption in nonimmune tissues is reduced to provide nutrients for the immune system [4]. We and others have also found a strong increase in the expression of the trehalose transporter *Tret1-1* and *Treh* in hemocytes during infection [4,8,9]. This suggests that trehalose is likely an important carbohydrate source for privileged immune cells.

Analysis of transcriptional changes at single-cell resolution suggested that larval hemocytes use lipids as the primary source to fuel the TCA cycle in the uninfected state [8]. However, half of the plasmatocytes in all clusters express an established glucose/trehalose transporter *MFS3* [21] as well as a putative monosaccharide transporter *sut1* [9], suggesting that plasmatocytes also utilize glucose. While plasmatocytes may consume lipids during infection, as deduced from gene expression, lamellocytes strongly up-regulate expression of several carbohydrate transporters and appear to mainly rely on saccharides as a source [8]. Our bulk transcriptomic data agree with these findings, and $^{13}$C-labeled carbohydrate tracing clearly supports these gene expression-based conclusions, showing that plasmatocytes from uninfected larvae do indeed metabolize glucose with a significant fraction metabolized via PPP.

The infection-induced carbohydrate consumption by hemocytes is mediated by a marked increase in the expression of 3 additional carbohydrate transporters besides MFS3 and sut1.

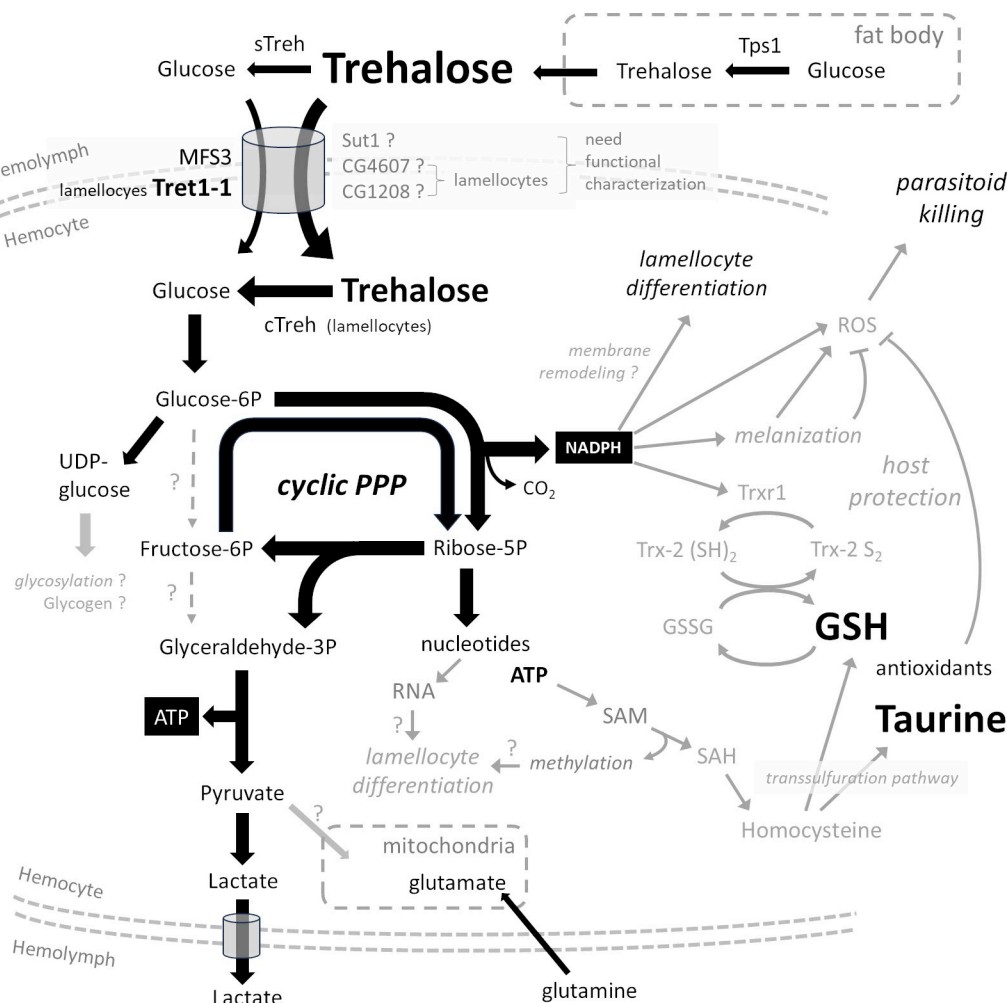

**Fig 7. Hemocyte metabolism during parasitoid infection and its links to immune processes.** A schematic model of hemocyte metabolism during parasitoid wasp infection. Metabolites, metabolic reactions/pathways, and processes marked in black have been examined in this study. Possible links to other processes and pathways (discussed in the main text) appear in gray. cTreh, cytoplasmic trehalase; GSH, reduced glutathione; GSSG, oxidized glutathione; PPP, pentose phosphate pathway; SAH, S-adenosylhomocysteine; SAM, S-adenosylmethionine; sTreh, secreted trehalase.

Like MFS3, Tret1-1 is an established glucose/trehalose transporter [19]. The mild reduction in lamellocyte abundance in the *Tret1-1 MFS3* double-null mutant indicates that at least one of the other, yet uncharacterized carbohydrate transporters (sut1, CG4607, or CG1208) contributes to glucose transport in hemocytes. Arguably, increased carbohydrate supply during immune challenge may be so critical as to require multiple, functionally redundant, transporters (Fig 7). This redundancy in carbohydrate transporters implies that loss of 1 or 2 transporter genes may not have a serious impact on immune response. However, the importance of carbohydrate supply to hemocytes is evident as hemocyte-specific silencing of oxidative PPP by *Zw* RNAi leads to loss of resistance. We have previously described the significance of a switch in systemic carbohydrate metabolism during immune response. Here, we show that hemocytes indeed require carbohydrates for an effective response in order to fuel PPP.

The different ways in which glucose is metabolized depend on the actual needs of cells (Fig 7 and [14]). If a cell primarily requires ATP, glucose is converted via glycolysis to pyruvate, which can further be metabolized in the mitochondria to produce multiple ATPs. If a cell

mainly requires nucleotides (typically a proliferating cell), glucose is metabolized by nonoxidative PPP to form ribose-5P (bypassing NADPH production by oxidative PPP), which is further utilized in de novo synthesis of purines/pyrimidines. If a cell primarily needs NADPH, glucose is metabolized in cyclic PPP, where the ribose-5P produced by oxidative PPP is recycled back to G6P, which can enter further rounds of oxidation in PPP. The cell can combine these pathways to obtain the optimal ratios of ATP, NADPH, pyruvate, and pentoses.

Our $^{13}$C tracing shows that hemocytes from uninfected larvae predominantly metabolize glucose and very little trehalose. When infected, hemocytes increase glucose metabolism and additionally metabolize trehalose. Expression of cTreh indicates that only differentiated lamellocytes cleave trehalose and likely also metabolize glucose, as MFS3 and Tret1-1 transport both trehalose and glucose [19,21]. Thus, by tracing $^{13}$C-labeled trehalose, we can specifically monitor lamellocyte metabolism. It is important to note that we observe metabolism of a heterogeneous pool of hemocyte types, and some hemocytes may use only some of the pathways in which we see $^{13}$C labeling.

As summarized in the overall model (Fig 7), hemocytes, including lamellocytes, increase their rate of glycolysis and lactate production during infection. A significant amount of glucose is shunted into the PPP, both in the absence of and during infection. G6P becomes partially labeled by $^{13}$C, indicating that a significant fraction of hemocytes, including lamellocytes, use cyclic PPP to re-oxidize G6P and maximize the yield of NADPH per glucose molecule [3]. A portion of G6P is returned to glycolysis at the glyceraldehyde-3P step, ultimately producing lactate. Some of the ribose-5P is diverted to de novo nucleotide synthesis. Thus, hemocytes, especially lamellocytes, couple NADPH production via cyclic PPP with de novo nucleotide synthesis and ATP production via downstream glycolysis. In addition, the specific pattern of partial labeling of pentoses suggests that some hemocytes generate pentoses via nonoxidative PPP regardless of infection. We could not determine how much carbohydrates were used in mitochondrial metabolism; we observed that hemocytes use amino acids as a source for the TCA cycle, or possibly for cytoplasmic citrate and lipid synthesis similarly in uninfected and infected larvae.

$^{13}$C tracing and impaired resistance in *Zw*-deficient larvae show the importance of cyclic PPP in immune cells. Although the significance of NADPH production in activated immune cells, particularly in conjunction with oxidative burst, is well known [34], we are aware of only one study showing the importance of cyclic PPP in immunity, specifically in mammalian neutrophils [3]. By providing evidence for the key role of cyclic PPP in invertebrate immune cells, our findings highlight its evolutionary significance. NADPH generated by cyclic PPP may serve a variety of purposes in immunity [34]. *Zw* null and RNAi animals show that oxidative/cyclic PPP is important for lamellocyte differentiation and for resistance (Fig 7). It is likely that NADPH is used during lamellocyte differentiation for reductive biosynthesis. For example, lamellocytes undergo significant changes in size, morphology, and adhesion capacity that distinguish them from their precursors [35]. Therefore, the use of NADPH in reductive biosynthesis of fatty acids can be hypothesized to play a key role in the remodeling of cell membranes during lamellocyte differentiation. In addition, there is a global shift in gene expression [8,9] and thus a requirement for RNA synthesis. Consequently, ribose-5P generated by PPP enters de novo nucleotide synthesis, as we observed by $^{13}$C labeling of AMP/ADP/ATP. De novo synthesized ATP is also required for S-adenosylmethionine production and increased methylation in activated immune cells [36]. Methylation of newly synthesized molecules (e.g., nascent proteins) may be important for lamellocyte differentiation but homocysteine, a product of methylation, is also used in the trans-sulfuration pathway [37] to produce the antioxidants glutathione and taurine (see below).

Suppression of oxidative/cyclic PPP affects resistance. NADPH is needed to produce ROS and these are needed to kill the parasitoid. NADPH oxidase reduces oxygen to superoxide anions, which are subsequently dismutated to hydrogen peroxide [38]. Interestingly, Nappi and Vass detected $H_2O_2$ in plasmatocytes, eventually attached to the egg, but not in lamellocytes [33]. This suggests that plasmatocytes use cyclic PPP to generate NADPH and $H_2O_2$, although our work does not exclude other sources of NADPH, such as isocitrate dehydrogenase and malic enzyme. Hydrogen peroxide can serve as a signal for further immune stimulation [39,40] or react with nitric oxide to form the hydroxyl radical, a very potent ROS [38]. Lamellocytes expressing the prophenoloxidase PPO3 and crystal cells expressing PPO2 [41] produce additional toxic molecules associated with melanization, which again requires NADPH [12]. Lamellocyte-mediated encapsulation and melanization concentrate the toxic reaction within the capsule, which is crucial for resistance. Thus, NADPH produced by oxidative PPP may be required for resistance due to reductive biosynthesis during lamellocyte differentiation and encapsulation and due to the production of toxic molecules inside the capsule. To kill the parasitoid, the metabolic activities of different hemocytes need to be coordinated. Resolving these roles requires time-controlled genetic manipulation of cyclic PPP, ROS production, and melanization cascade specific to the cell type in combination with, for example, genetically encoded metabolic sensors. Such manipulations would make the *Drosophila*-parasitoid model an excellent tool for investigating the role of cyclic PPP in immunity.

Trehalose is specifically metabolized by lamellocytes as they express cytoplasmic trehalase. The compensatory expression of alternative transcripts in a mutant lacking cTreh raises the question of why the capacity to metabolize trehalose is so important in lamellocytes. While systemic metabolism of trehalose is necessary for efficient lamellocyte differentiation, metabolism of trehalose within differentiating hemocytes is not. Thus, trehalose in the circulation appears to be important for maintaining adequate glucose levels for differentiating hemocytes [7]. Based on the expression of cTreh, trehalose is only metabolized by fully differentiated lamellocytes via cyclic PPP, generating NADPH. Survival rate of individuals with almost half of their hemocytes mutant for *Treh* is rather increased compared to controls. Thus, trehalose metabolism in hemocytes does not appear to be important for resistance. Encapsulation and melanization are thought to play a role not only in killing the pathogen but also in protecting the host from its own toxic immune reaction [12,13]. Based on our data, we propose that trehalose metabolism in lamellocytes may play a specific role in protecting the host: (1) Trehalose is metabolized by cyclic PPP, which generates NADPH. NADPH is required for the reduction of GSSG to GSH, which we found increased in hemocytes competent to respond at 22 hpi. Most hemocytes highly express genes of the thioredoxin system, but based on scRNAseq studies [8,9], lamellocytes show an even stronger expression. The thioredoxin system produces antioxidants including GSH. Depletion of trehalase activity from half of the lamellocytes led to increased hemolymph $H_2O_2$ levels at 30 hpi, strongly suggesting that trehalose metabolism is involved in ROS scavenging; down-regulation of this role may lead to a more toxic reaction for both the pathogen and the host. (2) Larvae with half of their hemocytes deficient in *Treh* are more resistant than controls but surviving adults are less fit. Molina-Cruz and colleagues showed that mosquito strains with higher ROS levels survived bacterial and *Plasmodium* infections at a higher rate, whereas dietary antioxidant supplementation reduced resistance [42]. Interestingly, the same antioxidants also significantly improved age-related loss of fecundity in mosquitoes [43]. It is important to emphasize that our study does not provide a direct causal link between increased ROS during the larval immune response and the reduced adult fitness. (3) *Treh* knockdown induced in the cTreh expression pattern increased pathogen killing, albeit at the expense of host survival as more flies died as pupae. Here, all tissues expressing cTreh were affected, including those that may normally contribute to host protection, resulting in

more severe effects. In conclusion, both the induction of $Treh^{-/-}$ mutant hemocyte clones and *Treh* RNAi increased resistance but impaired host survival and/or fitness, strongly suggesting that trehalose metabolism is needed for protecting the host from its own toxic immune response.

Although the intracellular metabolism of trehalose is restricted to lamellocytes and appears to be associated with an antioxidant response, this does not mean that these cells exclusively play a role in host protection, they may still mediate resistance. By encapsulating the parasitoid, lamellocytes are likely essential for containing the toxic reaction near the parasitoid, but they may also produce both ROS and antioxidants, which is not uncommon in immune cells [44]. Further studies should focus on lamellocytes to address the relationship between trehalose metabolism in cyclic PPP and the concurrent response to the parasitoid and its redox neutralization by the host.

In summary, an effective immune response to parasitoid wasp infection requires rapid and coordinated hemocyte activity, which includes lamellocyte differentiation. Additionally, capsule formation around the parasitoid egg is required, associated with a melanization cascade, and thus production of toxic molecules within the capsule. Lastly, protection of the host cavity from this toxic reaction is required for host survival. All these actions require changes in carbohydrate metabolism in hemocytes (Fig 7). Here, we show that systemic trehalose metabolism, including synthesis by Tps1 and conversion to glucose by Treh, is essential for adequate carbohydrate supply to hemocytes during infection, for lamellocyte differentiation and for host resistance. Nutrient supply to hemocytes is ensured by the expression of several carbohydrate transporters. While glucose is generally metabolized by hemocytes, trehalose is specifically metabolized only within lamellocytes by cytoplasmic trehalase. We demonstrate that both glucose and trehalose are metabolized via PPP, cyclic PPP in particular, which oxidizes G6P in multiple rounds to maximize NADPH production. PPP also connects to downstream glycolysis, which produces ATP and ends with the release of lactate. PPP and its links to several metabolic pathways support various activities required in the response to parasitoids. (1) We have shown that oxidative PPP is required for lamellocyte differentiation, implicating a role for NADPH in reductive biosynthesis, for example, in membrane remodeling. Differentiation could also be promoted by ribose-5P associated with nucleotide synthesis, which is required for broad changes in gene expression (new RNA and methylation). (2) We also demonstrated that oxidative PPP is required for resistance, which likely involves NADPH in the melanization cascade and in ROS production, both required for pathogen killing. (3) We observed increased production of the antioxidants glutathione and taurine, which requires NADPH/Trxr1-mediated reduction of GSSG to GSH and could also be promoted by coupling PPP-produced ribose-5P to the ATP-SAM-homocysteine-trans-sulfuration pathway. Antioxidants could explain the observed effects on the reduced fitness of trehalase knockout in hemocyte clones. Nevertheless, the potential link between sugars metabolized in PPP and host protective mechanisms requires further work, such as manipulating the thioredoxin system specifically in fully differentiated lamellocytes and studying the effects on larvae as well as development and physiology of surviving animals. It is difficult to separate resistance mechanisms from host protection when, for example, NADPH produced by cyclic PPP appears to be required for both. We were initially surprised that although trehalose metabolism seems to be important in hemocytes, we did not observe any effect on resistance. However, from an evolutionary perspective, protecting the host from its own immune response is no less important and probably no less energetically demanding than fighting the pathogen. The trade-off between survival and reproductive fitness may be of great evolutionary importance.

## Materials and methods

### Fly strains and cultivation

*Drosophila melanogaster* strain $w^{1118}$ (FBal0018186) in Canton S genetic background (FBst0064349) was used as a control line unless otherwise stated. Strains $Pgd^{n39}\ pn^1\ Zw^{lo2a}$ (FBst0006033), UAS-Zw-RNAi $Zw^{HMC03068}$ (FBal0292280), UAS-Tret1-1-RNAi $Tret1\text{-}1^{HMS02573}$ (FBal0281575), UAS-Treh-RNAi $Treh^{HMC03381}$ (FBal0292531) and control lines for RNAi $y^1\ v^1;\ P\{CaryP\}attP2$ (FBst0036303), $y^1\ v^1;\ P\{CaryP\}Msp300attP40$ (FBst0036304) and P{VALIUM20-EGFP.RNAi.1}attP40 (FBst0041555) were obtained from the Bloomington Drosophila Stock Center and UAS-Treh-RNAi $P\{GD5118\}v30731$ (FBst0458630) from Vienna Drosophila Resource Center. Strains $Treh^{c1}$ (FBal0321693), $Treh^{cs1}$ (FBal0321690), $Tps1^{MI03087}$ (FBal0260512) and $Tps1^{d2}$ (FBal0302039) were obtained from T. Nishimura, $MFS3^{CRISPR}$ (FBal0366542) and $Tret1\text{-}1^{XCVI}$ (FBal0319692) were obtained from S. Schirmeier. The *MSNF9-GFP* strain (*P{msn-GFP.F9}*; FBal0256873) was obtained from Viktor Honti. The *SrpD-Gal4* strain (FBtp0020112) was obtained from M. Crozatier, backcrossed into the $w^{1118}$ background, and recombined with P{tubP-GAL80ts}2 (FBti0027797), which was also backcrossed into $w^{1118}$ background, to generate the $w^{1118};\ +/+;\ SrpD\text{-}Gal4$ P{tubP-GAL80ts}2 line with Gal4 expression in all hemocytes but very low expression in the fat body at 25˚C (expression in the fat body is only present at 29˚C in this line). Line $w^{1118};\ P\{ry[+t7.2] = neoFRT\}42D$, $Treh^{cs1},\ P\{w[+mC] = Ubi\text{-}mRFP.nls\}2R\ /\ CyO;\ SrpD\text{-}Gal4\ /\ TM6B$ was generated by recombination of $P\{ry[+t7.2] = neoFRT\}42D\ P\{w[+mC] = Ubi\text{-}mRFP.nls\}2R$ (FBst0035496) with $Treh^{cs1}$ and by crossing to *SrpD-Gal4*. Line $w^{1118};\ P\{ry[+t7.2] = neoFRT\}42D,\ P\{w[+mC] = Ubi\text{-}GFP.nls\}2R1\ P\{Ubi\text{-}GFP.nls\}2R2\ /\ CyO;\ P\{y[+t7.7]\ w[+mC] = 20XUAS\text{-}FLPD5.PEST\}attP2\ /\ TM6B$ was generated by recombination of $P\{FRT(whs)\}G13\ P\{Ubi\text{-}GFP.nls\}2R1\ P\{Ubi\text{-}GFP.nls\}2R2$ (FBst0005826) with $P\{ry[+t7.2] = neoFRT\}42D$ (FBti0141188) and by crossing to $P\{y[+t7.7]\ w[+mC] = 20XUAS\text{-}FLPD5.PEST\}attP2$ (FBti0161054). All flies were grown on cornmeal medium (8% cornmeal, 5% glucose, 4% yeast, 1% agar, 0.16% methylparaben) at 25˚C.

### Generation of *Treh[RAΔG4]* mutant

CRISPR-mediated mutagenesis was performed by WellGenetics using modified methods of Kondo and Ueda [45]. In brief, the gRNA sequence TGATTGCTCGATGGATTCGC[TGG] was cloned into U6 promoter plasmid. Cassette *attP-Gal4-3xP3-RFP*, which contains attP, Gal4, RBS, and a floxed 3xP3-RFP, and 2 homology arms were cloned into pUC57-Kan as donor template for repair. *Treh*-targeting gRNAs and hs-Cas9 were supplied in DNA plasmids, together with donor plasmid for microinjection into embryos of control strain $w^{1118}$. F1 flies carrying selection marker of 3xP3-RFP were further validated by genomic PCR and sequencing. CRISPR generates a 47-bp deletion allele of *Treh* and is replaced by cassette attP-Gal4-3xP3-RFP (Fig 1). The line is depicted here as $Treh^{RAΔG4}$. $Treh^{RAΔG4}$ was 10 times backcrossed to our control $w^{1118}$ genetic background.

### Parasitoid wasp infection

Parasitoid wasps *Leptopilina boulardi* were reared on sugar agar medium (6% sucrose, 1.5% agar, 0.75% methylparaben) and grown by infection of wild-type *Drosophila* larvae. Early third instar larvae (72 h after egg laying) were infected with parasitoid wasps (= time point 0 hours). Weak infection (1 to 2 eggs per larva) was used for resistance and survival analysis. Strong infection (4 to 8 eggs per larva) was used for the rest of the experiments to obtain a strong and more uniform immune response. Infections were performed on 60-mm Petri dishes with

standard cornmeal medium for 15 min with periodic interruption of infecting wasps for weak infection and 45 min for strong infection.

## Hemocyte counting

Hemocytes were obtained from larvae by cuticle tearing of 1 larva in 15 µl PBS and counted based on morphology in Neubauer hemocytometer (Brand GMBH) using differential interference contrast microscopy.

## Resistance, survival, and fitness analysis

To determine survival and parasitoid resistance rates, infected/control larvae were placed in fresh vials (typical 1 experiment = 30 larvae/vial and 3 vials/genotype in 3 independent biological replicates of infection). To determine resistance, pupae were dissected 4 days after infection to count melanized wasp eggs (winning host kills pathogen) or surviving wasp larvae (winning parasitoid). For the survival experiment, emerged adult flies were counted as survivors of infection excluding flies without melanized capsule (if no melanized capsule was visible in the abdomen, the fly was dissected). Adult wasps that emerged from pupae were counted as adult winning parasitoids. The lifespan of surviving infected flies was determined by transferring flies to a fresh vial (10 female flies per vial) every 2 to 3 days and counting the days until death. Fitness was determined by leaving at least 5 wild-type uninfected males with at least 5 females that survived the infection (maximum 10 females per vial) in a vial, transferring them to a fresh vial every 2 to 3 days, and counting the number of offspring pupae throughout their lifetime.

## Gene expression analysis

Hemolymph was collected by rupturing 50 larvae in 2 µl of PBS on a microscope slide on ice, the hemolymph was transferred into 200 µl of Trizol reagent (Ambion), homogenized using a plastic motorized pestle, incubated for 15 min at room temperature and either frozen at −80°C or directly followed by RNA isolation using a Direct-zol RNA microprep kit (Zymo Research) according to the manufacturer's protocol. Reverse transcription was performed using Prime-Script reverse transcriptase (Takara) and gene expression was analyzed using the TP SYBR 2x mastermix (TopBio) with the primers listed in Table 1 on a CFX 1000 Touch real time cycler

**Table 1. Primers.**

| Gene (FlyBase ID) | Primer | Sequence 5′-3′ |
| --- | --- | --- |
| *Treh* (FBgn0003748) | cTreh-F | CGAGCAATCACAAAATGAACGG |
| | sTreh-F | CGACTATAACAATGCCATTCCCG |
| | Treh-R | CTGATTCTTGGCCTCCATCATG |
| | Treh-F1 | CAATCATTCCCGTGCCAAATC |
| | Treh-R1 | CCACGTACGACTTGACCATAC |
| | Treh-RB-F | CTGGTGCACAAAACAATACAGAT |
| | Treh-R2 | TTTGGATGGTGTGCAGCAGATT |
| *Tret1-1* (FBgn0050035) | Tret-RA-F | ACAAACTTCCCGAGGAAAACCT |
| | Tret-RA-R | CACACGATGATAGCCCAGCT |
| | Tret-RB-F | CACCGCGATGAAGATCCTGA |
| | Tret-RB-R | TGATGCCACCAACCCAAGAA |
| *Zw* (FBgn0004057) | Zw-F | GATAGCATCAAGGAGCAGTGT |
| | Zw-R | GCCTTGTTCTTGTTCTCCATAATC |
| *RpL32* (FBgn0002626) | RpL32-F | AAGCTGTCGCACAAATGGCG |
| | RpL32-R | GCACGTTGTGCACCAGGAAC |

(BioRad). Expression of a specific gene in each sample was normalized to expression of *RpL32* (FBgn0002626).

## Bulk RNAseq analysis

RNA was extracted from circulating hemocytes (72 h after egg laying = time of infection = 0 h, 81 h after egg laying = 9 h post infection/hpi and 90 h after egg laying = 18 hpi), from lymph glands (9 and 18 hpi) and from wing discs (9 hpi) of uninfected and infected third instar *w1118* larvae. Circulating hemocytes were obtained by ripping 100 larvae in ice-cold PBS directly into 1.5 ml centrifuge Eppendorf tubes, centrifuging 5 min at 360xg, removing the supernatant, and isolating RNA using Trizol reagent (200 μl) (Ambion) according to the manufacturer's protocol. Lymph glands and wing discs were dissected from larvae in ice-cold PBS, transferred to 1.5 ml centrifuge Eppendorf tubes with Trizol (200 μl) reagent (Ambion), homogenized using a plastic motorized pestle, followed by Direct-zol RNA microprep kit (Zymo Research) according to protocol. Frozen total RNA samples were sent to the Genomics Core Facility (EMBL Heidelberg) for preparation of barcoded 3′-end seq forward libraries, followed by deep uni-directional sequencing of 75-base long reads using Illumina NextSeq. Raw data are available at The European Nucleotide Archive under study accession number: PRJEB74490 (secondary acc: ERP159178) (https://www.ebi.ac.uk/ena/browser/view/PRJEB74490). Trimmed reads in Fastq files were mapped to the BDGP *Drosophila melanogaster* Release 6.29 genomic sequence using the Mapper for RNA Seq in Geneious prime software (Biomatters). Normalized counts of reads mapped to each gene annotation were calculated as TPM, expression levels were compared using the DESeq2 method in Geneious prime software, and data were exported to an Excel file (S1 Table).

## Metabolomics and stable $^{13}$C isotope tracing

$^{13}$C-labeled glucose feeding. $^{13}$C-labeled glucose (D-Glucose-$^{13}$C$_6$ isotope purity ≥99 atom % $^{13}$C - Sigma-Aldrich) was added to the semi-defined diet (per 100 ml: 0.62 g of agar, 8 g of brewer's yeast, 2 g of yeast extract, 2 g of peptone, 3 g of sucrose, 3 g of unlabeled glucose, 0.05 g of MgSO$_4$ × 6H$_2$O, 0.05 g of CaCl$_2$ × 2H$_2$O, 600 μl of propionic acid, 1 ml of 10% p-hydroxy-benzoic acid methyl ester in 95% ethanol). The diet mixture was brought to a boil, then cooled to 50 to 60°C with stirring and p-hydroxy-benzoic acid and propionic acid were added, and 2 ml of medium was taken into a Falcon tube and 100 μl of $^{13}$C-labeled glucose (600 mg/ml) was added (50% of the glucose in the diet was labeled). Approximately 1 ml of diet was poured into a glass vial, and 100 uninfected or infected larvae (16 h after the start of infection) were placed in the vial for 6 h. The larvae were then removed from the vial, washed twice in water and once in PBS and placed on a microscope slide covered with parafilm, and the PBS residue was removed with filter paper, and 4 μl of PBS was added to the larvae and each larva was ruptured, 20 μl of hemolymph was collected and transferred to sterile 1.5 ml Eppendorf polypropylene centrifuge tubes with 60 μl of PBS. Larvae were washed with an additional 25 μl of fresh PBS and 20 μl was recovered in the same tubes. Samples were centrifuged for 5 min at 360xg, and 95 μl of supernatant (with extracellular metabolites) was removed, 380 μl of cold acetonitrile-methanol (1:1) extraction buffer with addition of the internal standard 4-fluorophenylalanine (50 ng/sample) was added to control the extraction efficiency and mass spectrometer responses and samples were placed in liquid nitrogen. To collect metabolites from pelleted hemocytes, 50 μl of water was added, frozen in liquid nitrogen/thawed at 37°C 3 times (to disrupt the rigid cells of hemocytes), then 200 μl of cold acetonitrile-methanol (1:1) with addition of the internal standard 4-fluorophenylalanine (50 ng/sample) was added and stored at −80°C until LC-HRMS analysis.

Ex vivo hemocyte incubation with $^{13}$C-labeled glucose and trehalose. Larvae were washed first with distilled water and then with PBS to reduce contamination. Larval hemolymph was collected by carefully tearing the larvae on a glass microscope slide covered with parafilm. Hemolymph from 50 larvae was immediately collected into sterile 1.5 ml Eppendorf polypropylene centrifuge tubes prefilled with 100 µl PBS and centrifuged for 5 min at 25˚C, 360xg. The supernatant was then removed and the cells were mixed with labeled medium (5 mM $^{13}$C$_{12}$ labeled or unlabeled trehalose, 0.5 mM $^{13}$C$_6$ labeled or unlabeled glucose, 5 mM proline, 0.3 mM methionine, and 5 mM glutamine, all reagents from Sigma/Merck) supplemented with gentamicin (10 mg/ml; Gibco), amphotericin B (250 µg/ml; Gibco), and 0.1 mM phenylthiourea (PTU; Sigma/Merck) to prevent melanization. The hemocytes were then incubated for 40 min at 25˚C and 80% to 90% humidity. The cells were then centrifuged for 5 min at 25˚C, 360xg, the supernatant was removed, the cells were mixed with 50 µl of cold PBS and frozen in liquid nitrogen/thawed at 37˚C 3 times (to disrupt the rigid hemocyte cells). Finally, 200 µl of cold acetonitrile-methanol (1:1) was added, in addition, the internal standard 4-fluorophenylalanine (50 ng/sample) was added to control the extraction efficiency and mass spectrometer responses and samples were stored at −80˚C until LC-HRMS analysis.

Frozen samples were melted on ice, then samples were homogenized using a TissueLyser LT (Qiagen, Hilden, Germany) set to 50 Hz for 5 min (with a rotor pre-chilled to −20˚C). Homogenization and centrifugation (at 20,000×g for 5 min at 4˚C) was repeated twice and the 2 supernatants were combined. Post-centrifugation, the supernatant was collected and filtered using a 0.2 µm PVDF mini-spin filter (sourced from HPST, Prague, Czech Republic). This filtration process was carried out at 7,000 RPM and a temperature of 5˚C for 5 min. The filtered supernatant was then completely dried using a refrigerated vacuum concentrator (Jouan RC 10.10 and RCT 60, Saint Herblain, France). The dried aliquots were reconstituted with 50% acetonitrile (50 µl) and thoroughly mixed for 30 s before being placed in the ultrasonic bath for an additional 5 min. The prepared sample was then measured using LC-HRMS.

## LC/HRMS condition and data mining

In this study, the flux metabolomic analysis utilized the high-resolution mass spectrometer Q Exactive Plus combined with Dionex Ultimate 3000 and Dionex open autosampler from Thermo Fisher Scientific, San Jose, California, United States of America. The mass spectrometer was operated in negative (NESI) ion mode. Full MS scan mode was employed for detection, with a mass range of 70 to 1,050 Da. The Q-Exactive settings included a 70,000 resolving power (scan rate at ±3Hz), $3 \times 10^6$ automatic gain control (AGC) target, and a maximum ion injection time (IT) of 100 ms. The ionization parameters were set as follows: (+/−) 3,000 kV spray voltage, 350˚C capillary temperature, sheath gas at 60 arbitrary units (au), aux gas at 20 au, spare gas at 1 au, probe temperature 350˚C, and S-Lens level at 60 au. For accurate mass, lock masses of 301.9981 were utilized.

Chromatographic separation occurred on the SeQuant ZIC-pHILIC column (150 mm × 4.6 mm i.d., 5 µm, Merck KGaA, Darmstadt, Germany) with a flow rate of 450 µl/min, an injection volume of 5 µl, a column temperature of 35˚C, and a mobile phase gradient of 20% B at 0 min, 80% B at 20 min, 95% B at 20.1 min, holding at 95% B until 23.3 min, returning to 20% B at 23.4 min, and maintaining 20% B until 30.0 min. Here, A represented acetonitrile and B represented 20 mmol/L aqueous ammonium carbonate (pH = 9.2; adjusted by NH$_4$OH). The data were acquired and processed using homemade software MetaboliteMapper and Xcalibur software version 2.1 from Thermo Fisher Scientific, San Jose, California, USA. Raw data are available at figshare (https://doi.org/10.6084/m9.figshare.25525657.v1) [46]. All reported metabolites were identified using standards, except FGAR, UMP, UDP-galactose,

and 2PG, which were identified using online sources based on accurate mass and MS spectra. All satellites M+1, M+2, and M+3 was compensated for their natural representation. Full details on the identification and compensation of the natural occurrence can be found in S2 Table. For peak area analysis, the data were normalized to the total content of all screened unlabeled metabolites—the peak area of the metabolite in a particular sample was divided by the peak area of the same metabolite of the selected reference sample and this procedure was repeated for each individual unlabeled metabolite. These ratios of all metabolites in one particular sample were averaged to determine a normalization factor. We then divided the measured peak area by the normalization factor for that sample to obtain the normalized peak area values (normalization factors and normalized values are reported in S2 Table).

### Generation of hemocyte mutant clones by mitotic recombination

The *FRT RFP; SrpD-Gal4* control line (*w$^{1118}$; P{ry[+t7.2] = neoFRT}42D, P{w[+mC] = Ubi-mRFP.nls}2R / CyO; SrpD-Gal4 / TM6B*) or the *FRT Treh$^{cs1}$ RFP; SrpD-Gal4* mutant line (*w$^{1118}$; P{ry[+t7.2] = neoFRT}42D, Treh$^{cs1}$, P{w[+mC] = Ubi-mRFP.nls}2R / CyO; SrpD-Gal4 / TM6B*) were crossed with either flippase-free *FRT GFP* line (*w$^{1118}$; P{ry[+t7.2] = neoFRT}42D, P{w[+mC] = Ubi-GFP.nls}2R1 P{Ubi-GFP.nls}2R2 / CyO)* as control without clone induction or with *FRT GFP; UAS-Flp* line (*w$^{1118}$; P{ry[+t7.2] = neoFRT}42D, P{w[+mC] = Ubi-GFP.nls}2R1 P{Ubi-GFP.nls}2R2 / CyO; P{y[+t7.7] w[+mC] = 20XUAS-FLPD5.PEST}attP2 / TM6B*) to induce mitotic clonal recombination in hemocytes by expressing flippase using *SrpD-Gal4* driver. Larvae with ubiquitous red and green fluorescence, i.e., without balancers, were selected and dissected in PBS on a microscope slide to obtain hemocytes. Images of hemocytes were taken using red and green fluorescence and differential interference contrast microscopy. Merged images were used to count green, red, and heterozygous yellow hemocytes.

### Hemolymph hydrogen peroxide measurement

Early third instar larvae were selected and transferred to fresh medium in a Petri dish and were infected with parasitoid wasps. Larvae were collected from the Petri dish at 30 hpi, washed twice in water and once in PBS, placed on a microscope slide covered with parafilm, and the PBS residue was removed with filter paper. Hemolymph was obtained from the larvae by tearing the cuticle from 50 larvae and the hemolymph was collected in a sterile Eppendorf tube. The sample volume was adjusted to 50 μl with PBS. Pairs of samples were processed at the same time and the amount of $H_2O_2$ was immediately assessed using a colorimetric Hydrogen Peroxide Assay Kit (Abcam, Ab102500) according to the original protocol. The samples were not deproteinized. A fresh set of standards was used for each measurement.

### Immunohistochemistry

The central nervous system of infected and noninfected third instar larvae were dissected and stained according to standard protocols. The following primary antibodies were used: GFP anti-chicken (Abcam, 1:1,000), Rabbit anti-Rumpel (1:500; [47]), Elav anti-rat and Repo anti-mouse (Developmental Studies Hybridoma Bank, 1:5), Tret1-1 anti-guinea pig [1:50; [48]]. All secondary antibodies conjugated to Alexa Fluor 488, Alexa Fluor 568, or Alexa Fluor 647 were used at a ratio of 1:1,000 (Thermo Fisher Scientific). Confocal images were obtained using a Zeiss LSM 880 (Zeiss, Oberkochen, Germany) and analyzed using Fiji [49].

## Lipid droplet staining

Three third instar larvae were cleaned in PBS and placed in a 5 µl drop of PBS on a coverslip. Forceps were used to rip the cuticle, allowing hemolymph to flow out. Carcasses were discarded and hemolymph and left for 10 min to adhere to the coverslip. Cells were washed 3 × 5 min in 1× PBS and fixed in 4% paraformaldehyde (PFA) for 15 min. PFA was removed and cells were washed 3 × 5 min in 1× PBS then incubated with BODOPY 493/503 for 1 h at RT. Cells underwent a final washing step of 3 × 5 min in 1× PBS. A drop of Vectashield (vector) was added to the coverslip before being placed on a microscope slide. Edges were sealed with nail polish and images were captured on a FV3000 Olympus Confocal Microscope, using a 100× objective and 2× digital zoom.

## Data analysis

Data were analyzed and graphed using GraphPad Prism (GraphPad Software), with specific statistical tests shown in the legend of each figure.

## Supporting information

**S1 Fig. Cytoplasmic trehalase expression in hemocytes and wing imaginal disc.** *Treh [RAΔG4]* with a knocked-in Gal4 in the *Treh-RA* transcriptional variant drives UAS-GFP expression in the cytoplasmic trehalase expression pattern (cTreh>GFP). (A, B) Differential interference contrast (DIC) combined with fluorescence microscopy using 20× objective. (A) Hemocytes from uninfected third instar larvae with no expression of cTreh>GFP. (B) Hemocytes from larvae 22 h after wasp infection—while large flat lamellocytes express cTreh>GFP, no expression was detected in both round and spread plasmatocytes. (C) Parasitoid egg encapsulated by lamellocytes expressing cTreh>GFP 24 h after infection—DIC (left), green fluorescence (middle), and merged (right) image from a Leica Thunder Imaging Systems microscope using 20× objective. (D, E) Wing imaginal disc expressing cTreh>GFP similarly in uninfected (D) and infected (E) larvae. (D) DIC (left), green fluorescence (middle), and merged (right) image using 20× objective. (E) Merged image only. (TIF)

**S2 Fig. Cytoplasmic trehalase is expressed in glial cells of the central nervous system.** *Treh [RAΔG4]* with a knocked-in Gal4 in the *Treh-RA* transcriptional variant drives UAS-GFP expression in the cytoplasmic trehalase expression pattern (cTreh>GFP). (A–C and E–G) are single focal plane images that show cTreh>GFP (green) expression, Rumpel (magenta) predominantly expressed in ensheathing glia cells and Elav (blue) a neuronal specific marker. (D and H) are maximum projections highlighting the expression of cTreh>GFP (green). cTreh>GFP (green) shows overlap with Rumpel (magenta) but not Elav (blue), suggesting that the cytoplasmic trehalase is expressed in ensheathing glia, but not neurons. (I–K and M–O) are single confocal sections, cTreh>GFP (green), Repo (magenta) expressed in all glial nuclei and Tret1-1 (blue) expressed in perineurial glia, the outermost glial cell layer of the blood–brain barrier. (L and P) show a Z projection of larval brains with cTreh>GFP (green) and Repo (magenta) staining. There is overlap in expression of cTreh>GFP (green) and Repo (magenta). There is no evidence of cTreh>GFP (green) expression in perineurial glia (blue). (A–D and I–L) are brains of uninfected third instar larvae. (E–H and M–P) are third instar larval brains of infected animals. (A, C–E, G–I, K–M, O, P) show an overview of the central nervous system using 20× objective. (B, F, J, N) show a close up of the ventral nerve cord using 63× objective. (TIF)

**S3 Fig. Cytoplasmic trehalase expression in lamellocytes.** *Treh[RAΔG4]*-driven *UAS-mCherry* expression (red) was combined with lamellocyte-specific marker *MSNF9-GFP* (green) and hemocytes were analyzed at 18, 24, and 40 h postinfection (hpi) using differential interference contrast (DIC) combined with fluorescence microscopy using a 20× objective. *MSNF9-GFP* marker is already present at 18 hpi (top left, green labeled cells are not yet fully differentiated lamellocytes, some are attached to the parasitoid egg), while *Treh[RAΔG4]> UAS-mCherry* starts to appear at 24 hpi (top right; not all cells morphologically resembling lamellocytes express one or the other marker at this time point). Partially melanized parasitoid egg at 40 hpi (DIC image bottom left) is covered by lamellocytes, all of which express both *MSNF9-GFP* and *Treh[RAΔG4]>UAS-mCherry* (bottom right).
(TIF)

**S4 Fig. Analysis of hemocyte metabolism by stable $^{13}$C isotope tracing in vivo.** $^{13}$C labeling of metabolites in hemolymph and hemocytes obtained from larvae fed in vivo with fully labeled D-glucose-$^{13}$C$_6$ for 6 h starting at 16 h postinfection. (A, C) Hemolymph glucose and trehalose levels (peak area)—the bars show the mean metabolite amount expressed by the normalized peak area—stacked columns show unlabeled (gray), fully labeled (red) and partially labeled ($^{13}$C$_6$—pink in case of trehalose) parts; percentages above the columns express the labeled fractions. Samples were obtained from hemocytes of uninfected (Uninf) or infected (INF) larvae. (B) Labeled fractions of hemolymph glucose from uninfected (light gray) and infected (dark gray) larvae; graph is zoomed to m+1. . .m+6 fractions, m+0 is outside the graph area; bars represent means of 5 biological replicates ± SEM, each dot represents 1 biological replicate. (D, F) Intracellular hemocyte glucose and glucose-6P levels (peak area)—graphed in the same way as in (A). (E, G–I) Labeled fractions of intracellular hemocyte glucose (E), glucose-6P (G), ribulose-5P/ribose-5P (H), and sedoheptulose-7P (I) from uninfected (light gray) and infected (dark gray) larvae; graphed in the same way as in (B). (J) Intracellular hemocyte and circulating hemolymph total lactate levels (peak area) from uninfected (light gray) and infected (dark gray) larvae. Bars represent mean ± SEM, each dot represents 1 biological replicate, asterisk represents a significant difference between uninfected and infected samples tested by unpaired one-tailed Welch's *t* test. (K, L) Schematic examples of $^{13}$C (pink) isotope labeling of sedoheptulose-7P from fully labeled fructose-6P and unlabeled glyceraldehyde-3P resulting in an m+4 fraction (K) or from fully labeled ribose-5P unlabeled xylulose-5P resulting in an m+5 fraction (L)—details in S5 and S6 Fig. Numerical values are available in S1 Data.
(TIF)

**S5 Fig. Scheme of $^{13}$C isotope labeling of metabolites in the cyclic pentose phosphate pathway.** The cyclic pentose phosphate pathway (PPP) can recycle 6 pentoses—5C in black boxes (ribose-5P or xylulose-5P), which are formed from glucose-6-phosphate by oxidative PPP (OxPPP), into 4 hexoses—6C in blue boxes (glucose-6Ps) and 2 trioses—3C in blue boxes (glyceraldehyde-3Ps) by using transketolase and transaldolase. The recycled glucose-6Ps can enter further rounds of cyclic PPP to maximize NADPH production. Glyceraldehyde-3Ps can re-enter glycolysis. Metabolizing fully labeled glucose-$^{13}$C$_6$ in cyclic PPP produces partially labeled glucose-6P. Initially, when labeled metabolites begin to enter cellular metabolism and represent a minority fraction, the most common intermediate in cyclic PPP (and also in glycolysis) is fully labeled glyceraldehyde-3P-$^{13}$C$_3$, which combines with unlabeled metabolites to form glucose-6P-$^{13}$C$_3$ partially labeled at 3 carbons (pink highlight). Later, when more labeled metabolites enter cellular metabolism, the most common product of cyclic PPP is glucose-6P-$^{13}$C$_2$ (3 of 7 possible combinations after the first round, as highlighted in yellow). Red

circles represent $^{13}$C carbons and gray circles represent $^{12}$C carbons.
(TIF)

**S6 Fig. Scheme of $^{13}$C isotope labeling of metabolites in the nonoxidative pentose phosphate pathway.** The nonoxidative pentose phosphate pathway (PPP) produces ribose-5P from the glycolytic products fructose-6P and glyceraldehyde-3P by using transketolase and transaldolase. Metabolism of fully labeled glucose-$^{13}$C$_6$ in nonoxidative PPP produces mostly partially labeled ribose-5P (with $^{13}$C$_4$/m+4 being least likely) and less fully labeled ribose-5P-$^{13}$C$_5$. Red circles represent $^{13}$C carbons, gray circles $^{12}$C carbons, orange rectangles represent labeling in ribulose-5P, and gray rectangles in ribose-5P.
(TIF)

**S7 Fig. Analysis of hemocyte mitochondrial and amino acids metabolism by stable $^{13}$C isotope tracing in vivo.** $^{13}$C labeling of glycolytic and tricarboxylic acid cycle (TCA) metabolites and amino acids in hemolymph (above dashed line) and hemocytes (below dashed line) obtained from uninfected (light gray) and infected (dark gray) larvae fed in vivo with fully labeled D-glucose-$^{13}$C$_6$ for 6 h starting at 16 h postinfection. All graphs are zoomed to 13C-labeled fractions, m+0 is outside the graph area; bars represent means of 5 biological replicates ± SEM, each dot represents 1 biological replicate, numerical values are available in S1 Data.
(TIF)

**S8 Fig. Analysis of hemocyte de novo purine synthesis by stable $^{13}$C isotope tracing in vivo.** $^{13}$C labeling of ribulose-5P/ribose-5P, glycine and AMP in hemocytes obtained from uninfected (light gray) and infected (dark gray) larvae fed in vivo with fully labeled D-glucose-$^{13}$C$_6$ for 6 h starting at 16 h postinfection. All graphs are zoomed to $^{13}$C-labeled fractions, m+0 is outside the graph area; bars represent means of 5 biological replicates ± SEM, each dot represents 1 biological replicate. Fractions from uninfected and infected samples were compared using unpaired Welch's $t$ test with Holm–Sidak correction for multiple comparisons, asterisks label significant differences, ns labels nonsignificant differences. Levels of unlabeled AMP, ADP, and ATP (normalized peak areas) were compared using unpaired Welch's $t$ test; asterisks indicate $p$-value (** $P < 0.01$, **** $P < 0.0001$). Numerical values are available in S1 Data.
(TIF)

**S9 Fig. Lipid droplet composition in hemocytes.** Hemocytes of control (*Srp> P{y[+t7.7] = CaryP}attP2*) and hemocyte-specific RNAi of *Zw* induced by Srp-Gal4 (*Srp>P{TRiP. HMC03068}attP2*) third instar larvae were imaged and the lipid droplets were dyed using BODIPY 493/503. (A and D) Hemocytes of uninfected larvae show multiple small lipid droplets distributed through the cytoplasm of the cell. (B) Differentiating hemocytes of control animals 12 hpi contain many small lipid droplets, whereas (E) Zw knockdown hemocytes have fewer, larger lipid droplets. (C) Hemocytes of control larvae at 22 hpi have numerous smaller lipid droplets with some larger lipid droplets. (F) Srp > Zw RNAi hemocytes appear to have no visible lipid droplets at 22 hpi.
(TIF)

**S10 Fig. Effects of cytoplasmic trehalase-specific mutations.** (A) Map of the *trehalase* gene with individual transcripts (RA-RG) and sequence from RA first exons depicting wild-type, *Treh$^{c1}$* and *Treh$^{RAΔG4}$* mutations. *Treh$^{c1}$* deletes 20 bp including the first start codon. *Treh$^{RAΔG4}$* deletes 47 bp removing both start codons, which is replaced by a cassette containing the Gal4 coding sequence. Lines show introns, boxes show exons with coding sequence in orange. (B, C) Heat map of $^{13}$C-labeled fraction of metabolites from control and *Treh$^{c1}$* (B) and *Treh$^{RAΔG4}$*

(C) hemocytes in uninfected (Uninf) and infected (INF) conditions incubated ex vivo with labeled α,α–trehalose-$^{13}C_{12}$. (D) Number of lamellocytes 22 h after beginning of infection in control (*w$^{1118}$*) and in the *Treh$^{RAΔG4}$* mutant. Each dot represents number of lamellocytes in 1 larva, line represents mean, samples were compared by unpaired *t* test. Numerical values are available in S1 Data.
(TIF)

**S11 Fig. Hemocytes from control larvae unable to induce mitotic recombination in hemocytes.** Hemocytes from uninfected larvae that were unable to generate clones by mitotic recombination in hemocytes due to the lack of flippase (*FRT42D GFP / FRT42D Treh[cs1] RFP; Srp-Gal4 / +*)—all hemocytes express both GFP and RFP markers (yellow when merged). These larvae served as controls for larvae with mitotic recombination clones in hemocytes (Fig 6). Differential interference contrast (DIC) and fluorescence microscopy using a 40× objective.
(TIF)

**S12 Fig. Thioredoxin system in hemocytes.** (A) Scheme of the thioredoxin system in *Drosophila*. The reduction of the disulfide thioredoxin Trx $S_2$ to the reduced dithiol form Trx $(SH)_2$ is catalyzed by NADPH-dependent thioredoxin reductase (Trxr). Thioredoxin reduces glutathione disulfide (GSSG) to glutathione (GSH), an antioxidant that scavenges radicals via glutathione peroxidase (Gpx). *Drosophila* thioredoxin Trx-2 may also be a substrate for thioredoxin peroxidases (peroxiredoxins, Prx) that detoxify peroxides. (B) Bulk RNAseq of genes of the thioredoxin system expressed in circulating hemocytes from uninfected (Uninf, light gray bars) and infected (INF, dark gray bars) larvae 18 h after the start of infection. Hemocytes express thioredoxin reductase *Trxr1*, thioredoxin *Trx-2*, and various putative peroxiredoxins and glutathione peroxidases. Expressions are shown in transcripts per million (TPM), bars represent mean values; dots represent biological replicates, error bars represent ± SEM. (C) Single-cell RNAseq plot of *Trxr1* expression in hemocytes from wasp-infected larvae for 48 h, obtained from the single-cell RNA-seq data portal of DRSC/Perrimon lab (https://www.flyrnai.org/scRNA/), showing stronger expression in lamellocytes. *Atilla* (lamellocyte marker) and *Hml* (plasmatocyte marker) expression is shown for comparison. (D) Graph of *Trx-2* expression in hemocytes based on single-cell RNA-seq data portal (https://www.flyrnai.org/tools/single_cell/web/) showing that a higher percentage of lamellocytes express *Trx-2* more strongly than other hemocytes. Numerical values are available in S1 Data and in S1 Table.
(TIF)

**S13 Fig. Hydrogen peroxide levels in hemolymph and lifespan of female survivors.** (A) $H_2O_2$ levels in hemolymph of uninfected (light gray) and infected (dark gray) control larvae at 30 hpi. (B) $H_2O_2$ levels in hemolymph from uninfected control (light gray) and uninfected larvae with *Treh[cs1]* mutant clones (recombination, purple) at time corresponding to 30 hpi. Dots represent paired biological replicates (infection performed and $H_2O_2$ measured in the same time for compared samples); data were compared using two-tailed paired *t* test, ns . . . not significant. (C) Lifespan of control males and males with *Treh[cs1]* mutant clones surviving infection was tested by Gehan–Breslow–Wilcoxon test, median survival 45 days for control and 39 days for males with *Treh[cs1]* mutant clones (*P* = 0.489). (D) Lifespan of control females and females with *Treh[cs1]* mutant clones surviving infection was tested by Gehan–Breslow–Wilcoxon test, median survival 48 days for control and 34 days for females with *Treh[cs1]* mutant clones (*P* = 0.0237). Numerical values are available in S1 Data.
(TIF)

**S14 Fig. Resistance upon trehalase knockdown using *Treh[RAΔG4]* knock-in Gal4.** (A–F) Survival of parasitoid wasp infection after Treh-RNAi induced by Gal4 driver knocked in cTreh (*Treh[RAΔG4]*). Two RNAi lines were used—*Treh*[HMC03381] (purple in A–C) and *P {GD5118}*[v30731] (blue in D–F). Larvae with *CyoGFP* balancer instead of *Treh[RAΔG4]*, i.e., without Gal4 and RNAi induction, from the same cross and infected in the same cage were used as control (green dots in A–F), resulting in paired data (connected by line). (A, D) Paired dots show the percentage of surviving flies; (B, E) show the percentage of developing parasitoids; (C, F) show the percentage of developing flies that die as pupae; dots represent biological replicates; data were compared using two-tailed paired *t* test, asterisks indicate *p* value (*$P < 0.05$, ** $P < 0.01$). (G, H) *Treh* expression analyzed by RT-qPCR 22 h after the start of infection measured by expression of the common region for all *Treh* transcripts using Treh-F1/Treh-R1 primers; bars show fold change compared to uninfected +/CyoGFP; EGFP-RNAi samples from larvae carrying *CyoGFP* balancer and uninduced *EGFP-RNAi* construct—green in (G); expression levels were normalized by *RpL32* expression in each sample, each dot represents a biological replicate. An unpaired one-tailed Welch's *t* test was used to compare samples; *$P < 0.05$, ** $P < 0.01$. Due to scale, only uninfected samples are shown in (G); both uninfected (uninf) and infected (INF) samples are shown in (H). Heterozygous *Treh[RAΔG4]* mutation (+/cTreh Gal4 inducing control EGFP RNAi; gray bar) reduces *Treh* expression to approx. 60% in both uninfected and infected larvae. cTreh Gal4 (heterozygous mutation)-induced Treh-RNAi further reduces *Treh* expression to 30% (uninfected-G), 44% (infected Treh-RNAi-GD5118, blue) and 15% (infected Treh-RNAi-HMC03381, purple)–(H). (I) No effect of the *CyoGFP* balancer (green) used to select control larvae (without *Treh[RAΔG4]* knock-in Gal4) from RNAi larvae after infection was detected compared to larvae without balancer (gray). FLY—percentage of surviving flies; WASP—percentage of adult parasitoid wasps. Error bars represent ± SEM. Numerical values are available in S1 Data.
(TIF)

**S1 Table. Gene expression analysis by bulk RNAseq of circulating hemocytes, lymph gland, and wing disc during parasitoid wasp infection.** MS Excel sheets with gene expression in circulating hemocytes (first sheet), lymph gland, and wing disc (second sheet). RNA was extracted 72 h after egg laying = time of infection = 0 h, 81 h after egg laying = 9 h postinfection/hpi and 90 h after egg laying = 18 hpi), from hemocytes, lymph glands, and wing discs of the third instar *w*[1118] larvae. Barcoded 3′-end seq forward libraries were subjected to deep unidirectional sequencing of 75-base long reads using Illumina NextSeq. Trimmed reads in Fastq files were mapped to the BDGP *Drosophila melanogaster* release 6.29 genomic sequence (gene names correspond to this release) using the Mapper for RNA Seq in Geneious prime software (Biomatters). Normalized counts of reads mapped to each gene annotation were calculated as transcripts per million (TPM), expression levels were compared using the DESeq2 method in Geneious prime software. Raw data are available at The European Nucleotide Archive under Study accession number: PRJEB74490 (secondary acc: ERP159178) (https://www.ebi.ac.uk/ena/browser/view/PRJEB74490).
(XLSX)

**S2 Table. Metabolomics and stable [13]C isotope tracing in circulating hemocytes during parasitoid wasp infection.** MS Excel sheets with stable [13]C isotope tracing experiments. List of metabolites, their characterizations, identification, HPLC/HRMS parameters, and compensation of natural occurrence is on the first sheet [S2a_List of Metabolites]. Data from the following experiments are in individual sheets (values are raw or normalized areas under respective chromatographic peaks with compensation of natural occurrence in yellow-highlighted cells marked by * in headings): [S2b_13C in vivo]—raw and normalized data from hemocytes

obtained 22 h after start of infection from uninfected and infected $w^{1118}$ larvae fed a diet with 50% D-Glucose-$^{13}C_6$ for the last 6 h. [S2c_13C ex vivo] raw and normalized data from hemocytes obtained 22 h after start of infection from uninfected and infected *Srp>P{y[+t7.7]* = *CaryP}attP2* control larvae and incubated for 40 min in medium containing 5 mM unlabeled trehalose and 0.5 mM $^{13}C_6$ labeled glucose or 5 mM $^{13}C_{12}$ labeled trehalose and 0.5 mM unlabeled glucose. [S2d_13C ex vivo Trehcs1], [S2e_13C ex vivo Trehc1], and [S2f_13C ex vivo TrehRAdG4] sheets—raw data from hemocytes obtained 22 h after start of infection from uninfected and infected $w^{1118}$ control and *Treh$^{cs1}$*, *Treh$^{c1}$*, or *Treh$^{RA\Delta G4}$* mutant larvae and incubated for 40 min in medium containing 5 mM $^{13}C_{12}$ labeled trehalose and 0.5 mM unlabeled glucose. [S2g_13C ex vivo Srp-Tret1-1-RNAi] raw and normalized data from hemocytes obtained 22 h after start of infection from uninfected and infected *Srp>P{y[+t7.7]* = *CaryP} attP2* control larvae and larvae with hemocyte-specific Tret1-1 RNAi *Srp>P{TRiP.HMS02573} attP2* and incubated for 40 min in medium containing 5 mM $^{13}C_{12}$ labeled trehalose and 0.5 mM unlabeled glucose. Raw data are available at figshare ([https://doi.org/10.6084/m9.figshare.25525657.v1](https://doi.org/10.6084/m9.figshare.25525657.v1)).
(XLSX)

**S1 File. Carbohydrate transport and metabolism gene expression analysis by bulk and single-cell transcriptomics.** Table with bulk RNAseq gene expressions (transcripts per million—TPM, average values) of genes from SLC2 and SLC17 family of sugar transporters in *Drosophila*—the intensity of the red color corresponds to the TPM value. Bulk RNAseq expressions of selected genes in bar graphs—each dot represents a biological replicate in TPM, bars represent mean ± SEM. Tret1-1 gene map and transcript-specific expression analysis. Single cell-RNAseq of circulating hemocytes—dot plots with average gene expressions in hemocyte clusters; color gradient of the dot represents the expression level, the size represents percentage of cells expressing the gene per cluster; downloaded from [www.flyrnai.org/tools/single_cell/web/](www.flyrnai.org/tools/single_cell/web/). Single-cell RNAseq of circulating hemocytes—t-Distributed Stochastic Neighbor Embedding (t-SNE) plots of Harmony-based batch correction of wasp infected 48 h data sets downloaded from [www.flyrnai.org/scRNA/blood/](www.flyrnai.org/scRNA/blood/) (for comparison, plots with plasmatocytes marker *Hml* and lamellocyte marker *Atilla* are shown).
(PDF)

**S2 File. Glycolytic and pentose phosphate pathway gene expression analysis by bulk and single-cell RNAseq.** Diagram showing metabolic pathways and tables with gene expression corresponding to [Fig 2](Fig 2). Table with bulk RNAseq gene expressions (transcripts per million—TPM, average values) of glycolytic and PPP genes in *Drosophila*—the intensity of the red color corresponds to the TPM value. Expression of selected genes in bulk RNAseq (this work) shown as bar graphs (each dot represents a biological replicate in TPM, bars represent mean ± SEM) and single-cell RNAseq (downloaded from [www.flyrnai.org/scRNA/blood/](www.flyrnai.org/scRNA/blood/) and [www.flyrnai.org/tools/single_cell/web/](www.flyrnai.org/tools/single_cell/web/)) shown by dot plots with average gene expressions in hemocyte clusters (color gradient of the dot represents the expression level, the size represents percentage of cells expressing the gene per cluster) and t-Distributed Stochastic Neighbor Embedding (t-SNE) plots of Harmony-based batch correction of wasp infected 48 h data sets (for comparison, plots with plasmatocytes marker *Hml* and lamellocyte marker *Atilla* are shown).
(PDF)

**S1 Data. Raw data/numerical values for data presented in graphs.** The MS Excel file containing the data used to generate the plots in Figs [1](1), [3](3), [4](4), [5](5), [6](6), [S4](S4), [S7](S7), [S8](S8), [S10](S10), [S12](S12), [S13](S13) and [S14](S14). The data for each figure are shown on individual sheets of this file with the panel number

indicated next to the data.
(XLSX)

## Acknowledgments

We thank Takashi Nishimura, Stefanie Schirmeier, Michele Crozatier, Viktor Honti, and Bloomington Drosophila Stock Center for fly and wasp stocks. We thank to Lucie Hrádková for laboratory management and Marcela Jungwirthová for project management, and all members of Doležal and Šimek laboratories for their help with work. We thank Vladimír Beneš of Genomics Core Facility (EMBL Heidelberg, Germany) and especially Jan Provazník for RNA-seq services and WellGenetics Inc. (New Taipei City, Taiwan) for CRISPR-mediated mutagenesis. We thank Jason Tennessen for advice on metabolomics and Marek Jindra and Dalibor Kodrík for sharing reagents and equipment, and Marek Jindra also for his extensive help with editing the manuscript, as well as Rebecca Collier.

## Author Contributions

**Conceptualization:** Martin Moos, Tomáš Doležal.

**Data curation:** Martin Moos, Tomáš Doležal.

**Formal analysis:** Martin Moos, Tomáš Doležal.

**Funding acquisition:** Michalina Kazek, Tomáš Doležal.

**Investigation:** Michalina Kazek, Lenka Chodáková, Katharina Lehr, Pavla Nedbalová, Ellen McMullen, Adam Bajgar, Stanislav Opekar, Martin Moos, Tomáš Doležal.

**Methodology:** Michalina Kazek, Lukáš Strych, Pavla Nedbalová, Adam Bajgar, Stanislav Opekar, Petr Šimek, Martin Moos, Tomáš Doležal.

**Resources:** Stanislav Opekar, Petr Šimek, Martin Moos, Tomáš Doležal.

**Supervision:** Petr Šimek, Martin Moos, Tomáš Doležal.

**Validation:** Martin Moos.

**Visualization:** Michalina Kazek, Lenka Chodáková, Lukáš Strych, Ellen McMullen, Tomáš Doležal.

**Writing – original draft:** Michalina Kazek, Tomáš Doležal.

**Writing – review & editing:** Michalina Kazek, Lenka Chodáková, Ellen McMullen, Petr Šimek, Martin Moos, Tomáš Doležal.

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
