## [Editor Report · Decision Letter 0]

14 Aug 2023

Dear Dr Dolezal, 

Thank you for submitting your manuscript entitled "Metabolism of glucose and trehalose by cyclic pentose phosphate pathway is essential for effective immune response in Drosophila" for consideration as a Research Article by PLOS Biology. 

Your manuscript has now been evaluated by the PLOS Biology editorial staff as well as by an academic editor with relevant expertise and I am writing to let you know that we would like to send your submission out for external peer review. I should say that, while we are interested in your study, we have yet to make a firm call about whether the manuscript offers a sufficient advance for PLOS Biology - and so we will be looking for strong reviewer support in that regard. 

Before we can send your manuscript to reviewers, we need you to complete your submission by providing the metadata that is required for full assessment. To this end, please login to Editorial Manager where you will find the paper in the 'Submissions Needing Revisions' folder on your homepage. Please click 'Revise Submission' from the Action Links and complete all additional questions in the submission questionnaire.

Once your full submission is complete, your paper will undergo a series of checks in preparation for peer review. After your manuscript has passed the checks it will be sent out for review. To provide the metadata for your submission, please Login to Editorial Manager (https://www.editorialmanager.com/pbiology) within two working days, i.e. by Aug 16 2023 11:59PM.

Kind regards,

Lucas

Lucas Smith, Ph.D.

Senior Editor

PLOS Biology

lsmith@plos.org

---

## [Decision Letter · Decision Letter 1]

11 Oct 2023

Dear Dr Dolezal,

Thank you for your patience while your manuscript "Metabolism of glucose and trehalose by cyclic pentose phosphate pathway is essential for effective immune response in Drosophila" was peer-reviewed at PLOS Biology. Please accept my apologies for the time it has taken to send you a decision. In this case, for some reason, it took us a bit longer than normal to secure a full set of reviewers (it can be a bit tricky to find reviewers around the start of the school year). Your study has now been evaluated by the PLOS Biology editors, an Academic Editor with relevant expertise, and by several independent reviewers.

In light of the reviews, which you will find at the end of this email, we would like to invite you to revise the work to thoroughly address the reviewers' reports.

As you will see below, the reviewers find the study interesting, but report that further experimental work is needed to strengthen the conclusions. The reviewers also highlight that the manuscript should be thoroughly edited to make the findings more accessible to a broad audience. Given that the isotope tracing of an insect immune response is one of the major advances in this paper, we would like to emphasize the need to thoroughly address, w/ new data and analyses, the technical concerns raised by Reviewers 1 and 2, regarding that data. We also agree with Reviewer 3's comment that a more direct manipulation is needed to show that intracellular trehalose protects against immune toxicity, and we think further experiments will be necessary to strengthen that conclusion. 

Given the extent of revision needed, we cannot make a decision about publication until we have seen the revised manuscript and your response to the reviewers' comments. Your revised manuscript is likely to be sent for further evaluation by all or a subset of the reviewers.

**IMPORTANT - SUBMITTING YOUR REVISION**

*Re-submission Checklist*

*Published Peer Review*

*PLOS Data Policy*

*Blot and Gel Data Policy*

Sincerely,

Luke

Lucas Smith, Ph.D.

Senior Editor

PLOS Biology

lsmith@plos.org

REVIEWS:

Reviewer #1, Zheng-Jiang Zhu (note, Reviewer 1 has signed this review): 

General comments: 

Kazek et al mainly used 13C tracing and genetic manipulation of glucose and trehalose metabolism to study changes in metabolism in Drosophila hemocytes. Authors showed results that hemocytes from uninfected larvae metabolize predominantly glucose and very little trehalose. When infected, hemocytes increase glucose metabolism and additionally metabolize trehalose. The manuscript was generally interesting. Concerns regarding data processing of 13C tracing data and conclusions drew from the results in this manuscript should be carefully addressed.

Major concerns:

1. 13C tracing data is fundamental to the conclusions claimed by authors. However, a basic and critical data processing procedure, that is natural isotope correction, was not conducted in the manuscript. The raw data was subjected to comparative analyses directly.

2. For the labeling data in Figure 3, as well as that in Fig S5 and Fig S6, the results are very likely to be different to that authors presented if labeling fractions are compared between uninfected and infected groups. Authors are encouraged to added analysis results based on labeling fractions and discuss them.

3. Authors used the LC-MS method as previously reported (https://doi.org/10.3390/metabo12020163). In the current manuscript, conclusions based on Ru5P and R5P should be carefully checked. I recommend authors showing the EICs and both Ru5P and R5P measured in hemocytes and chemical standards.

4. As results from line 303 to line 327, conclusions drew from ex vivo data were very different from that in in vivo. The explained reason regarding heterogeneous hemocyte types is not convinced since ex vivo experiments were conducted using a mixture of hemocytes as well. 

5. Ex vivo experiments were conducted by collecting hemocytes with bleeding larvae 22 hrs post infection. Are there any different impacts on metabolism by distinct infection time?

6. Despite stating in the title and abstract, data on immune responses was insufficient but with a few data of lamellocyte differentiation and resistance in Figure 4 in the manuscript. How about the impacts on releasing of different types of cytokines?

7. Stating "incorporation is the same" by analyzing peak area data is incorrect. Checking labeling extents between groups is reasonable. 

8. I recommend adding more basic and critical information on the metabolite identification in this manuscript. How to annotate the metabolites in this study (MS1, MS2, and RT)? What is the meaning of cutoff for annotation? Has a subset of metabolites been confirmed with authentic standards? How many metabolites were identified as MSI level 1, 2, and 3?

Reviewer #2, Tina Mukherjee (note, Reviewer 2 has signed this review): In this manuscript, Kazak et al investigates the metabolic changes in hemocytes during an immune response to parasitoid wasps. Previously, they have shown that during wasp infection, adenosine released from immune cells mediates a systemic metabolic switch leading to more carbohydrate availability to immune cells at the expense of other tissues. These findings led the authors to conduct a detailed investigation into the actual metabolic changes in the larval hemocytes in the same infection scenario. Here, they have combined bulk transcriptomics and 13C metabolic tracing in the hemocytes along with genetic manipulation of carbohydrate metabolism. Based on these experiments, the central message of this manuscript is that lamellocytes predominantly metabolize trehalose by cyclic pentose phospate pathway to not only provide resistance but perhaps to also protect the host from its own toxic response. The study makes an important contribution to the field of immunometabolism highlighting the critical role of cyclic pentose phosphate pathway in immunity against parasitoid wasps. The paper effectively utilises already published single cell RNA datasets to delineate the metabolic signatures of specific immune cell type. Further, while it would have been beneficial to have used positionally labelled glucose/trehalose in their metabolic flux analysis, the authors took an interesting and thorough approach to compare the fractions with different numbers of labeled carbons to highlight the involvement of cyclic PPP that will be beneficial to the scientific community to interpret their own flux data. 

Overall, the paper is interesting and makes an insightful approach to addressing metabolic rewiring during immune, however, I have a few concerns related to their metabolic analysis and the experimental evidence provided to say that metabolism of glucose and trehalose by cyclic PPP is essential for effective immune response at the current stage and addressing them will help clarify these concerns. I list them hereby:

1. Their transcriptome analysis from their previous work is contrasting to the results shown in this current study. The authors here show that the expression of glycolytic genes is down-regulated upon infection at 18h as shown on in Fig. 2, while the text describes it as no overall change as mentioned in the line 197-198, while in their previous work (Bajgar et al., 2015), they have shown that glycolytic gene expression is up-regulated upon infection. This variation in the results from their previous work and interpretation of data in the current work needs to be thoroughly addressed. 

2. Authors have used both in-vivo and ex-vivo 13C-glucose labelling to understand the role of metabolic pathways upon wasp-infection in hemocytes. This is one of the very few studies which has utilised metabolomics and 13C-label incorporation both in-vivo and ex-vivo approaches to understand such mechanisms. The authors are suggested to present their 13C label incorporation data as percentage label and the total incorporation graphs to be included in the supplementary data. This will make it easier for the reader to understand their findings as in the following instances (listed below) it is not sure what the comparison is with:

a. Line 222-223 "The increased 13C labelled glucose shows that hemocytes increase uptake of sugars upon infection (Fig 3C), again in agreement with our previous results" is not clear, whether the authors are comparing the label incorporation with uninfected hemocytes label or the 13C label in haemolymph upon infection.

b. In Fig. 3O, TCA cycle labelling is shown with C3 label in citrate from 13C6-glucose, as pyruvate from glucose contributes 2 carbons in citrate by conversion through PDH, the 13C2 label needs to be checked as well to comment on the contribution of TCA cycle.

c. Partial labelling of G6P is an important conclusion drawn from Fig. 3D and G, as mentioned in the line 364-365 similar increase in C1-5 labelling is not observed in the ex-vivo labelling, a discussion about the difference in labelling patterns in the ex-vivo and in-vivo conditions would be appreciated.

3. The data to support Trehalase function in mature lamellocytes needs to be strengthened. Except the one image of bled hemocytes to support cytoplasmic trehalase expression in fully differentiated lamellocytes, the auhtors need to convincingly support this result, as it is a major claim made by the authors. Antibody staining with marker like misshappen that marks matured lamellocytes need to be provided. Secondly, Treh RNAi in the hemocytes does not show any significant defect in lamellocytes differentiaton and immune response on their own needs stringent analysis. The authors have used Srp-gal4, which is also expressed in the fat body at 29 degrees for which the authors had kept the RNAi crosses at 25 degrees. Could the Srp>Treh RNAi not showing any effect on lamellocytes be due to Gal4 not reaching optimum activity at 25 degrees; also it is important to utilise multiple RNAi lines at this point and an additional Hemocyte driver (Hml>). Finally, the use of clones to dissect the autonomous role of treh metabolism in lamellocytes on improved resistance needs to reconsidered in light of the above experiments suggested. 

Minor comments:

1. Have the authors tested for normality of data distribution? If not, then instead of using unpaired t-test, Mann-Whitney or even t-test with Welch's correction can be used.

2. Since a large part of the work/discussion centers around lamellocyte-specific PPP having a host protective effect against toxic reactions, did the authors attempt to assess ROS levels or other readouts in control versus Treh mutant conditions?

3. Throughout the discussion, there is a major focus on the association of PPP-mediated NADPH production with nucleotide synthesis which might be important for lamellocyte differentiation. However, considering that NADPH also supports fatty acid synthesis, a very simple lipid staining in conjunction with plasmatocyte and lamellocyte marker can be done.

4. In the methods section the details for LCMS methods used should be provided and the Q1/Q3 and other MS parameters needs to be provided in the supplementary file.

Reviewer #3: In this manuscript, Kazek et al present a detailed evaluation of carbon metabolism in Drosophila melanogaster in support of an immune response against parasitoid wasps. Anti-parasitoid immune defense requires substantial energy and metabolic restructuring in support of hematopoesis and differentiation of lamellocytes. Kazek et al show that trehalose is taken up by hemocytes during an immune response, and cleaved into glucose to fuel the pentose phosphate pathway, in support of lamellocyte differentiation. The authors also find that differentiated lamellocytes make use of intracellular trehalose, although this is not required for resistance to parasitoids. They instead interpret that the utilization of intracellular trehalose may somehow be protective against immune autotoxicity, although this hypothesis is not directly tested or strongly supported by any data sh

---

## [Decision Letter · Decision Letter 2]

25 Mar 2024

Dear Dr Dolezal,

Thank you for your patience while we considered your revised manuscript "The role of glucose and trehalose metabolism by cyclic pentose phosphate pathway in pathogen resistance and host protection in Drosophila" for publication as a Research Article at PLOS Biology. This revised version of your manuscript has been evaluated by the PLOS Biology editors, the Academic Editor and the original reviewers.

Based on the reviews and on our Academic Editor's assessment of your revision, we are likely to accept this manuscript for publication, provided you satisfactorily address the remaining point raised by the Reviewer 1. Overall, we agree with this reviewer that the current study may not be able to clearly distinguish between R5P and Ru5p. We think this point should be addressed by changing the text to explain this limitation and by changing 'R5p' to 'R5p/Ru5p' or something similar in the text and figures. If you prefer to keep the claims specific to R5p, we think additional experiments and controls would be needed to more clearly demonstrate the specificity. If you would like to take on this work, we would be happy to extend the deadline for your revision. 

**IMPORTANT: In addition to addressing the reviewer comments, please also make sure to address the following data and other policy-related requests.

1) TITLE: We think the title could be strengthened by more clearly conveying the specific findings of the study. If you agree (and if supported), we suggest it be changed to something like:

"Glucose and trehalose metabolism through the cyclic pentose phosphate pathway shape pathogen resistance and host protection in Drosophila"

2) PRESENTATION: We note that reviewer 3 comments that the presentation of your study is a bit dense, and editorially we feel that the manuscript would benefit from another edit, focused on language and grammar, with the goal of making the study more accessible to our broad readership. If possible, we think it may be helpful to run your study by a critical colleague, and to even get input from someone not working directly in this field. Again - if you need a bit of extra time to do this, we are happy to extend the deadline for the revision. 

3) DATA: You may be aware of the PLOS Data Policy, which requires that all data be made available without restriction: http://journals.plos.org/plosbiology/s/data-availability. For more information, please also see this editorial: http://dx.doi.org/10.1371/journal.pbio.1001797

>> Please deposit all raw data for the RNA-seq and mass spec datasets generated here on a publicly available repository. 

4) DATA: For all of the other experiments presented in the study, we would not require the raw data. However, we need you to provide all individual quantitative observations that underlie the data summarized in the figures and results of your paper be made available in one of the following forms:

a. Supplementary files (e.g., excel). Please ensure that all data files are uploaded as 'Supporting Information' and are invariably referred to (in the manuscript, figure legends, and the Description field when uploading your files) using the following format verbatim: S1 Data, S2 Data, etc. Multiple panels of a single or even several figures can be included as multiple sheets in one excel file that is saved using exactly the following convention: S1_Data.xlsx (using an underscore).

b. Deposition in a publicly available repository. Please also provide the accession code or a reviewer link so that we may view your data before publication. 

>>Regardless of the method selected, please ensure that you provide the individual numerical values that underlie the summary data displayed in the following figure panels as they are essential for readers to assess your analysis and to reproduce it:

Fig1A-C,E-G; Fig 3A-H; Fig 4B-D; Fig 5A-G; Fig 6C-M;

Fig S4A-J; Fig S7; Fig S8; Fig S10B-D; Fig S12B; Fig S13A-C; Fig S14A-I;

>>Please also ensure that figure legends in your manuscript include information on where the underlying data can be found, and ensure your supplemental data file/s has a legend.

>>Please ensure that your Data Statement in the submission system accurately describes where your data can be found.

5) DATA NOT SHOWN: Please note that per journal policy, we do not allow the mention of "data not shown", "personal communication", "manuscript in preparation" or other references to data that is not publicly available or contained within this manuscript. I noticed one instance where data was discussed but not shown (line 606 - male lifespan...). Please provide a relevant figure panel presenting the results needed to support this statement. 

6) CODE: Per journal policy, if you have generated any custom code during the curse of this investigation, please make it available without restrictions upon publication. Please ensure that the code is sufficiently well documented and reusable, and that your Data Statement in the Editorial Manager submission system accurately describes where your code can be found. Please note that we cannot accept sole deposition of code in GitHub, as this could be changed after publication. However, you can archive this version of your publicly available GitHub code to Zenodo. Once you do this, it will generate a DOI number, which you will need to provide in the Data Accessibility Statement (you are welcome to also provide the GitHub access information). See the process for doing this here: https://docs.github.com/en/repositories/archiving-a-github-repository/referencing-and-citing-content

We expect to receive your revised manuscript within two weeks. 

*Published Peer Review History*

*Press*

Sincerely,

Luke

Lucas Smith, Ph.D.

Senior Editor

lsmith@plos.org

PLOS Biology

Reviewer remarks:

Reviewer #1: 

1. It is not true for the authors stated that "We are able to distinguish between Ru5P and R5P, but a small fraction of R5P may consist of Ru5P - this does not have a major impact on our interpretations". As can be clearly observed in Table S2a that the EICs Ru5P and R5P were highly overlapped, so these two metabolites can not be distinguished in samples. Thus, results and conclusions drew from R5P and Ru5P in this work are not convincible. 

2. It should be "METLIN" but not "METLYN" in table S2.

Reviewer #2, Tina Mukherjee (note, reviewer 2 has signed this review): The authors have addressed all concerns raised in their revised version. I congratulate the authors on this fine work that they have conducted. 

Reviewer #3: Kazek and colleagues have revised their manuscript in response to prior review, notably converting the presentation of their 13-C measurements to fractions as opposed to absolute peak areas, and in response to my major comment, adding a measurement of ROS activity in the hemolymph after infection and trehalase knockdown. The manuscript represents a tremendous amount of work and it is expertly performed. The presentation remains a bit dense, but the authors have done well with their summary paragraphs in each main results section. The figures included in this submission are not of sufficient resolution for publication, but I assume that is for review only and the final figures will be high resolution. My comments on the previous version of the manuscript have been adequately addressed and it is my impression that the comments of the other reviewers have also been sufficiently addressed.

---

## [Editor Report · Decision Letter 3]

12 Apr 2024

Dear Dr Dolezal,

Thank you for the submission of your revised Research Article "Glucose and trehalose metabolism through the cyclic pentose phosphate pathway shapes pathogen resistance and host protection in Drosophila" for publication in PLOS Biology, and thank you for addressing the last reviewer and editorial requests in this revision. On behalf of my colleagues and the Academic Editor, Alex P Gould, I am pleased to say that we can in principle accept your manuscript for publication, provided you address any remaining formatting and reporting issues. These will be detailed in an email you should receive within 2-3 business days from our colleagues in the journal operations team; no action is required from you until then. Please note that we will not be able to formally accept your manuscript and schedule it for publication until you have completed any requested changes.

PRESS

Sincerely, 

Luke

Lucas Smith, Ph.D.

Senior Editor

PLOS Biology

lsmith@plos.org